# A Mathematical Framework for AI-Human Integration in Work

**L. Elisa Celis** [1]  **Lingxiao Huang** [2]  **Nisheeth K. Vishnoi** [1]

## Abstract

The rapid rise of Generative AI (GenAI) tools has sparked debate over their role in complementing or replacing human workers across job contexts. We present a mathematical framework that models jobs, workers, and worker-job fit, introducing a novel decomposition of skills into decision-level and action-level subskills to reflect the complementary strengths of humans and GenAI. We analyze how changes in subskill abilities affect job success, identifying conditions for sharp transitions in success probability. We also establish sufficient conditions under which combining workers with complementary subskills significantly outperforms relying on a single worker. This explains phenomena such as *productivity compression*, where GenAI assistance yields larger gains for lower-skilled workers. We demonstrate the framework's practicality using data from O*NET and Big-bench Lite, aligning real-world data with our model via subskill-division methods. Our results highlight when and how GenAI complements human skills, rather than replacing them.

## 1. Introduction

The rapid emergence of capabilities in Generative Artificial Intelligence (GenAI) has drawn global attention. Multimodal models like OpenAI's GPT-4 and DeepMind's Gemini seamlessly interpret and generate text and images, transforming tasks such as content creation, summarization, and contextual understanding (OpenAI, 2023; DeepMind, 2023). Similarly, models like OpenAI's Codex and GitHub Copilot show strong performance in code generation and debugging (OpenAI, 2024; Jaffe et al., 2024). Notably, GPT-4 scores on standardized tests, including SAT, GRE, and AP, are comparable to those of human test-takers (OpenAI, 2023).

These advances have intensified focus on their implications for work and workers. Institutions such as the BBC (Annabelle Liang, 2024; Chris Vallance, 2023), the IMF (Cazzaniga et al.), and the World Economic Forum (Shine & Whiting, 2023), along with many researchers (Brynjolfsson et al., 2023; Felten et al., 2023; Noy & Zhang, 2023; Agarwal et al., 2023; Angelova et al., 2023; Cabrera et al., 2023; Vaccaro et al., 2024; Jaffe et al., 2024; Otis et al., 2024), have explored the evolving labor landscape. A growing view holds that GenAI may recompose, rather than eliminate, work. Autor (2024) argues that GenAI has the potential to enable middle-skill workers to take on tasks traditionally reserved for high-skill experts. Still, some see GenAI as a disruptive force: the IMF estimates that nearly 40% of jobs could be affected, raising concerns about large-scale displacement (Cazzaniga et al.; Chris Vallance, 2023; Microsoft, 2023; Paradis, 2024; Kochhar, 2023). Others emphasize complementarity: GenAI tools can enhance human capabilities rather than replace them (Acemoglu & Johnson, 2023). Empirical studies show that such tools improve the performance of less experienced workers, narrowing the productivity gap with more skilled professionals (Brynjolfsson et al., 2023; Noy & Zhang, 2023). This raises a central question: *Do GenAI tools substitute for human workers—or enable them to succeed in new ways?*

Recent studies have empirically examined GenAI's impact on various aspects of work, including accuracy, productivity, and implementation cost (Agarwal et al., 2023; Brynjolfsson et al., 2023; Vaccaro et al., 2024; Jaffe et al., 2024; Klubnikin et al., 2024; Brodsky, 2024; Anthropic, 2023; Guo et al., 2025). Vaccaro et al. (2024), for example, analyzed 106 experiments comparing human-AI collaboration to human- or AI-only performance on job tasks. They found that while collaboration improves outcomes in content creation, it lags in decision-making, underscoring the nuanced ways GenAI complements human skills. Brynjolfsson et al. (2023) and Jaffe et al. (2024) studied GenAI integration in real-world workflows, showing measurable productivity gains. In one case, AI-assisted customer service agents resolved 14% more issues per hour, suggesting that GenAI can amplify human efficiency. Other work has highlighted implementation barriers: studies such as Klubnikin et al. (2024); Brodsky (2024) identify hidden costs and operational challenges that hinder widespread GenAI adoption.

---

[1]Yale University, USA. [2]State Key Laboratory of Novel Software Technology, New Cornerstone Science Laboratory, Nanjing University, China. Correspondence to: Nisheeth K. Vishnoi <nisheeth.vishnoi@gmail.com>.

*Proceedings of the 42ⁿᵈ International Conference on Machine Learning*, Vancouver, Canada. PMLR 267, 2025. Copyright 2025 by the author(s).

This paper focuses on understanding the impact of GenAI on *job accuracy*. Addressing this requires modeling jobs, worker abilities (whether human or AI), and how these abilities relate to job performance. A key resource we draw on is the *Occupational Information Network (O\*NET)* (U.S. Department of Labor, Employment and Training Administration, 2023), a comprehensive database maintained by the U.S. Department of Labor that provides standardized descriptions of thousands of jobs. For example, O\*NET characterizes *Computer Programmers* by skills (e.g., *Programming*, *Written Comprehension*, *Oral Expression*), tasks (e.g., *Correcting errors*, *Developing websites*), and knowledge areas. Each skill is rated by importance and required proficiency—for instance, *Writing* might be rated 56/100 for importance and 46/100 for proficiency. While O\*NET offers rich and structured data, it does not specify how tasks depend on skills, nor how to evaluate performance at the level of a skill, task, or job (see Section E for more detail). Recent work has begun to address these limitations using compositional and task-skill dependency models (Arora & Goyal, 2023; Okawa et al., 2023; Yu et al., 2024).

Metrics for evaluating human workers include *Key Performance Indicators (KPIs)* (KPI.org, 2024), customer feedback, peer reviews, and productivity measures. For example, KPIs might assess a programmer's ability to fix a certain number of bugs within a set timeframe, while customer feedback evaluates the perceived quality of service. However, such metrics often conflate outcomes with underlying competencies, making it difficult to isolate a worker's ability on specific skills. In contrast, GenAI tools are typically evaluated using skill-specific benchmarks in areas such as coding, writing, and mathematical reasoning (Borji, 2022; Bubeck et al., 2023; bench authors, 2023; OpenAI, 2023; Abdin et al., 2024; Yu et al., 2024; Reid & Vempala, 2024; He et al., 2024). For instance, bench authors (2023) introduced the *BIG-bench Lite (BBL)* dataset, which evaluates 24 skills, including code generation, by comparing the performance of GenAI models and human workers (see Section E.2). One representative task, *Automatic Debugging*, tests whether a model can infer the state of a program given partial code—e.g., determining the value of a variable at a specific line without executing the program.

While GenAI tools perform well on structured or repetitive tasks, they often struggle with skills that require contextual understanding, planning, or emotional intelligence (Bender et al., 2021; Arora & Goyal, 2023; Services, 2024; Mahowald et al., 2024). These limitations are compounded by noisy and narrowly scoped evaluations, making it difficult to draw reliable conclusions about performance (Miller, 2024; bench authors, 2023). Consider a company that sets a KPI target of fixing 20 bugs per week. If a programmer fixes 18, their score would be $18/20 = 90\%$. But such metrics conflate distinct abilities: reasoning skills (e.g., diagnosing

the root cause) and action skills (e.g., implementing the fix). This conflation obscures the underlying sources of success or failure, leading to biased or incomplete evaluations. In addition, evaluations are rarely standardized across settings, and lab-based assessments often fail to capture the broader skillsets required in real-world jobs (Microsoft, 2023; Vaccaro et al., 2024). In summary, significant challenges remain in evaluating human workers and GenAI tools: (i) The conflation of reasoning and action, leading to inaccurate performance attributions. (ii) The statistical noise inherent in limited-scope evaluations. (iii) The lack of standardization, creating inconsistencies across assessments.

**Our contributions.** We introduce a mathematical framework to assess job accuracy by modeling jobs, workers, and success metrics. A key feature is the division of skills into two types of subskills: decision-level (problem solving) and action-level (solution execution) (Section 2). Skill difficulty is modeled on a continuum $[0, 1]$, where 0 represents the easiest and 1 the hardest. Workers, whether human or AI, are characterized by ability profiles $(\alpha_1, \alpha_2)$, representing decision- and action-level abilities. These profiles quantify a worker's capability for each skill $s \in [0, 1]$, incorporating variability through probability distributions. Jobs are modeled as collections of tasks, each requiring multiple skills. We define a *job-success probability* metric, combining error rates across skills and tasks to evaluate overall performance (Equation (1)). This framework addresses challenges in evaluation by isolating abilities, accounting for noise, and providing a metric for accuracy. Our main results include:

- *Phase transitions:* small changes in average ability can cause sharp jumps in job success (Theorem 3.2).

- *Merging benefit:* combining workers with complementary subskills yields superadditive gains (Theorem 3.3).

- *Compression effect:* our model explains the *productivity compression* observed by Brynjolfsson et al. (2023), where GenAI narrows the performance gap between low- and high-skilled workers (Corollary 3.4).

- *Empirical validation:* we apply our framework to real-world data from O\*NET and BIG-bench Lite, aligning task descriptions and GenAI evaluations with subskill-based models (Section 4).

- *Interventions and extensions:* we explore upskilling strategies through ability/noise interventions (Section D.1), and extend our analysis to dependent subskills and worker combinations (Figures 3, 4.2).

- *Bias and noise:* we analyze how misestimated ability profiles distort evaluations (Section D.2).

Our findings inform strategies for integrating GenAI into the workplace, including combining human and AI strengths, designing fairer evaluations, and supporting targeted upskilling. Additional related work is reviewed in Section A.

## 2. Model

**The job model.** We model a job as a collection of $m \geq 1$ tasks, where each task $T_i$ requires a subset of $n \geq 1$ skills. This induces a bipartite *task-skill dependency graph* where edges connect tasks to the skills they depend on (Figure 12), aligning with prior work (Arora & Goyal, 2023; Okawa et al., 2023; Yu et al., 2024).

Each skill $j$ is decomposed into two subskills: a *decision-level* component (e.g., problem-solving, diagnosis) and an *action-level* component (e.g., execution, implementation), following distinctions made in cognitive and labor models (Licklider, 1960; Kahneman, 2011; Inga et al., 2023). For example, the skill "programming" involves both solving a problem (decision-level) and implementing a solution in code (action-level). This decomposition allows for more precise modeling of ability, especially for evaluating hybrid human-AI work. Adapting from O*NET, each skill $j \in [n]$ is associated with subskill difficulties $s_{j1}, s_{j2} \in [0,1]$, where 0 indicates the easiest and 1 the hardest. These scores are used to index the worker's ability distributions. This representation mirrors the proficiency levels used in O*NET and simplifies the mathematical formulation; see Section E.

**The worker model.** We model a worker by two ability profiles, $\alpha_1$ and $\alpha_2$, which govern their decision-level and action-level subskills, respectively. Each profile maps a subskill difficulty $s \in [0,1]$ to a probability distribution over $[0,1]$, from which a performance value is drawn, representing the worker's effectiveness on that subskill. This reflects the stochastic nature of skill performance (Sadeeq, 2023; Bubeck et al., 2023). We consider ability profiles in which, for each subskill difficulty $s \in [0,1]$, the worker has an *average ability* $E(s)$, and their actual performance is modeled by adding a stochastic *noise* term $\varepsilon(s)$. The resulting ability value is used to define the worker's distribution over outcomes for that subskill.

We study two natural noise models: (i) *Uniform noise:* The noise term $\varepsilon(s)$ is sampled from a scaled uniform distribution: $\varepsilon(s) \sim \min\{E(s), 1 - E(s)\} \cdot \mathrm{Unif}[-\sigma, \sigma]$, where $\sigma \in [0,1]$ controls the noise level. The scaling ensures that the perturbed value remains within the valid range $[0,1]$. This model provides a simple yet effective way to introduce bounded variability and is often used in our analysis. (ii) *Truncated normal noise:* Here, we model $\varepsilon(s)$ using a truncated normal distribution: $\varepsilon(s) \sim \mathrm{TrunN}(E(s), \sigma^2; 0, 1)$, where the mean is $E(s)$, the variance is $\sigma^2$, and the support is clipped to remain within $[0,1]$. This model captures fluctuations consistent with human performance variability and is aligned with empirical measurements of GenAI tool behavior (bench authors, 2023) (see Figure 9).

We assume that the average ability function $E(s)$, which maps subskill difficulty $s \in [0,1]$ to expected performance,

is monotonically decreasing in $s$. That is, workers are not expected to perform worse on easier subskills ($s = 0$ denotes the easiest and $s = 1$ the hardest).

*Linear ability profile.* A natural used form is the *linear* function: $E(s) := c - (1 - a)s$, for parameters $a, c \in [0,1]$ satisfying $a + c \geq 1$. Here, $c$ represents the worker's maximum ability (attained at $s = 0$), and $1 - a$ is the rate at which ability decreases with difficulty. This profile aligns with evaluations of GenAI tools (Hendrycks et al., 2021; bench authors, 2023); see also Figure 11 and Section B.1 for Big-bench Lite analysis. As a special case, setting $a = 1$ yields a *constant* ability function $E(s) \equiv c$, where the worker has uniform performance across all subskills.

*Polynomial ability profile.* To model nonlinear improvements, we also consider the *polynomial* form: $E(s) = 1 - s^\beta$, where $\beta \geq 0$ controls the sensitivity of ability to difficulty. Larger values of $\beta$ produce sharper gains in ability as $s \to 0$, representing workers whose skills improve rapidly as tasks become easier.

Note that nearby subskills (e.g., $s = 0.7$ and $0.8$) yield similar values of $E(s)$, reflecting the smoothness of the ability profile and inducing implicit correlations across adjacent subskills. Section B.1 provides additional visualizations of these profiles under uniform noise. Section B.2 shows that these profiles satisfy *stochastic dominance* (Definition B.1): for any fixed $s \in [0,1]$ and threshold $x \in [0,1]$, the probability $\Pr_{X \sim \alpha(s)}[X \geq x]$ increases monotonically with the average ability.

**Measuring job-worker fit.** To evaluate how well a worker fits a job, we define a sequence of aggregation functions that compute error rates at the subskill, skill, task, and job levels, based on the worker's ability profiles $(\alpha_1, \alpha_2)$.

For each skill $j \in [n]$, let $s_{j1}, s_{j2} \in [0,1]$ denote the difficulty levels of its decision-level and action-level subskills. We define the random subskill error rate as: $\zeta_{j\ell} := 1 - X$, where $X \sim \alpha_\ell(s_{j\ell})$, for $\ell \in \{1,2\}$. That is, $\zeta_{j\ell}$ represents the probability of failure (or error rate) for the $\ell$-th subskill of skill $j$, drawn from the worker's ability distribution.

To compute the error rate for skill $j$, we apply a *skill error function* $h : [0,1]^2 \to [0,1]$, which aggregates the two subskill error rates $\zeta_{j1}$ and $\zeta_{j2}$. This gives the overall error rate for skill $j$, combining both decision-level and action-level performance. We assume a common skill error function $h$ for all skills, typically chosen as the average $h(a,b) = \frac{a+b}{2}$ or the maximum $h(a,b) = \max\{a,b\}$ (KPI.org, 2024; Walker, 2023), both of which are monotonic.

Similarly, to compute the error rate for a task $T_i$, we apply a *task error function* $g : [0,1]^* \to [0,1]$, which maps the error rates of the skills in $T_i$ to a task-level error: $g\left(\{h(\zeta_{j1}, \zeta_{j2})\}_{j \in T_i}\right)$.

Finally, we define a *job error function* $f$ : $[0,1]^m \rightarrow [0,1]$, which aggregates the task error rates into a single overall job error rate: $\text{Err}(\zeta) := f\left(g\left(\{h(\zeta_{j1}, \zeta_{j2})\}_{j \in T_1}\right), \ldots, g\left(\{h(\zeta_{j1}, \zeta_{j2})\}_{j \in T_m}\right)\right)$.

In our empirical analysis (Section 4), we instantiate $g$ and $f$ as weighted averages, where the weights reflect the importance of individual skills and tasks. More generally, we assume that $h$, $g$, and $f$ are monotonic: improving subskill abilities cannot increase the resulting error rate.

Given a threshold $\tau \in [0,1]$, we say a job *succeeds* if the job error rate satisfies $\text{Err}(\zeta) \leq \tau$. The *job success probability* for a worker with profiles $(\alpha_1, \alpha_2)$ is then defined as:

$$P(\alpha_1, \alpha_2, h, g, f, \tau) := \Pr_{\zeta_{j\ell}}\left[\text{Err}(\zeta) \leq \tau\right]. \quad (1)$$

In the special case of noise-free abilities, the job success probability becomes binary, taking values in $\{0, 1\}$.

## 3. Theoretical results

This section presents our theoretical results on how the job success probability $P$ (1) varies with worker ability parameters. We fix the job instance throughout: task-skill structure $\{T_i\}$, subskill difficulties $\{s_{j\ell}\}$, aggregation functions $h, g, f$, and success threshold $\tau$. Given this setup, $P(\alpha_1, \alpha_2, h, g, f, \tau)$, abbreviated as $P$, depends only on $\alpha_1$ and $\alpha_2$. We begin by analyzing a single worker with decision- and action-level profiles parameterized by average ability $\mu_\ell \geq 0$ and noise level $\sigma_\ell \geq 0$ for $\ell \in \{1, 2\}$. Fixing $\mu_2, \sigma_1, \sigma_2$, we show that $P$ undergoes a sharp *phase transition* as $\mu_1$ crosses a critical value (Theorem 3.2).

Next, we study the benefits of *merging* two workers with complementary abilities. Given workers A and B with profiles $(\alpha_1^{(A)}, \alpha_2^{(A)})$ and $(\alpha_1^{(B)}, \alpha_2^{(B)})$, we consider all four possible combinations of decision-level and action-level profiles: $(\alpha_1^{(A)}, \alpha_2^{(A)})$, $(\alpha_1^{(A)}, \alpha_2^{(B)})$, $(\alpha_1^{(B)}, \alpha_2^{(A)})$, $(\alpha_1^{(B)}, \alpha_2^{(B)})$. Theorem 3.3 gives conditions under which merging improves success probability. We conclude with Corollary 3.4, which connects this analysis to the *productivity compression* observed in Brynjolfsson et al. (2023).

### 3.1. Notation and assumptions

We begin with a structural observation: since the error aggregation functions $h, g, f$ are all monotonic, their composition Err is also monotonic in the subskill error rates $\zeta_{j\ell}$. Hence, to show that the job success probability $P$ increases with $\mu_1$, it suffices to show that higher $\mu_1$ leads to lower values of $\zeta_{j1}$. Recall that $\zeta_{j\ell} = 1 - X$, where $X \sim \alpha_\ell(s_{j\ell})$. Thus, lower error rates correspond to higher sampled ability values. This follows from stochastic dominance: for any $s, x \in [0,1]$, $\Pr_{X \sim \alpha_\ell(s)}[X \geq x]$ increases with $\mu_\ell$; see Proposition B.1.

**Independent noise assumption.** We assume independence across the random noise realizations at the subskill level. This assumption pertains only to *execution noise*: once the ability profiles $\alpha_1$ and $\alpha_2$ are fixed, the realized performances across subskills are modeled as independent draws. The ability profiles themselves may still induce correlations—e.g., via a smooth expected ability function $E(s)$ where adjacent subskills have similar mean performance.

**Assumption 3.1** (**Noise independence**). *For all $j \in [n]$ and $\ell \in \{1, 2\}$, the subskill error rates $\zeta_{j\ell}$ are independent drawn from $\alpha_\ell(s_{j\ell})$.*

This modeling choice is standard for both human and GenAI workers. For example, GenAI tools often produce conditionally independent task outputs given a fixed model state. We explore noise-dependent settings in Section 4 and extend our theoretical analysis to such settings in Section C.5. In particular, we find that strong correlations in noise reduce the sensitivity of $P$ to changes in ability parameters.

**Notation on sensitivity to ability parameters.** To study how the job success probability $P$ varies with ability, we begin by analyzing the expected job error rate: $\text{Err}_{\text{avg}}(\mu_1, \sigma_1, \mu_2, \sigma_2) := \mathbb{E}_\zeta[\text{Err}(\zeta)]$, where the expectation is taken over the subskill error rates $\zeta_{j\ell} = 1 - X$, with $X \sim \alpha_\ell(s_{j\ell})$. This quantity captures the average error rate for a fixed job instance and ability parameters $(\mu_1, \sigma_1), (\mu_2, \sigma_2)$. To quantify the impact of decision-level ability on average error, we consider $\left|\frac{\partial \text{Err}_{\text{avg}}}{\partial \mu_\ell}\right|$. This measures how sensitive the average job error is to changes in $\mu_\ell$, holding the other parameters fixed. Given fixed values of the noise levels and the other ability parameter, the minimum influence of $\mu_1$ on the expected job error is defined as:

$$\text{MinDer}_{\mu_1}(\sigma_1, \mu_2, \sigma_2) := \inf_{\mu_1 \geq 0}\left|\frac{\partial \text{Err}_{\text{avg}}}{\partial \mu_1}(\mu_1, \sigma_1, \mu_2, \sigma_2)\right|.$$

A large value of $\text{MinDer}_{\mu_1}$ indicates that small increases in decision-level ability $\mu_1$ can significantly reduce the expected job error. $\text{MinDer}_{\mu_2}(\mu_1, \sigma_1, \sigma_2)$ is defined similarly.

As an example, consider $\text{Err}(\zeta) = \frac{1}{2n}\sum_{j,\ell} \zeta_{j\ell}$ be average over all subskills and $\alpha_\ell(s) = 1 - (1 - a_\ell)s + \varepsilon_\ell(s)$ be linear profiles, where $\varepsilon_\ell(s) \sim \min\{(1 - a_\ell)s, 1 - (1 - a_\ell)s\} \cdot \text{Unif}[-\sigma_\ell, \sigma_\ell]$. We compute that $\left|\frac{\partial \text{Err}_{\text{avg}}}{\partial a_\ell}\right| = \frac{1}{2n}\sum_{j \in [n]} s_{j\ell}$. This implies that $\text{MinDer}_{\mu_1} = \frac{1}{2n}\sum_{j \in [n]} s_{j1}$ and $\text{MinDer}_{\mu_2} = \frac{1}{2n}\sum_{j \in [n]} s_{j2}$.

**Lipschitz assumption.** We assume the job error function Err is $L$-Lipschitz with respect to $\ell_1$-norm:

$$|\text{Err}(\zeta) - \text{Err}(\zeta')| \leq L \cdot \|\zeta - \zeta'\|_1 \quad \text{for all } \zeta, \zeta' \in [0,1]^{2n}.$$

When Err is the average of subskill errors, $L = \frac{1}{2n}$.

## 3.2. Threshold effect in job success probability

We now quantify how the success probability $P$ changes with decision-level ability $\mu_1$, holding other parameters fixed. We prove a sharp threshold behavior: once the expected job error crosses the success threshold $\tau$, even small changes in $\mu_1$ can cause the success probability to jump from near zero to near one. This phenomenon—formalized below—shows a *phase transition* in job success probability, controlled by a critical ability level $\mu_1^c$.

**Theorem 3.2 (Phase transition in job success probability).** *Fix the job instance, action-level ability $\mu_2$, and noise levels $\sigma_1, \sigma_2$. Let $\mu_1^c$ be the unique value such that the expected job error equals the success threshold:*

$$\mathsf{Err}_{avg}(\mu_1^c, \sigma_1, \mu_2, \sigma_2) = \tau.$$

*Let $\theta \in (0, 0.5)$ be a confidence level, and define the transition width: $\gamma_1 := \frac{L\sqrt{n(\sigma_1^2 + \sigma_2^2) \cdot \ln(1/\theta)}}{\mathrm{MinDer}_{\mu_1}(\sigma_1, \mu_2, \sigma_2)}$, where $L$ is the Lipschitz constant of the job error function. Then the job success probability satisfies:*

$$P \leq \theta \text{ if } \mu_1 \leq \mu_1^c - \gamma_1 \text{ and } P \geq 1 - \theta \text{ if } \mu_1 \geq \mu_1^c + \gamma_1.$$

Theorem 3.2 shows that increasing $\mu_1$ by approximately $2\gamma_1$ transitions the success probability $P$ from at most $\theta$ to at least $1 - \theta$. A smaller value of $\gamma_1$ implies that even modest gains in decision-level ability can have a significant impact on job success. Conversely, a slight increase in the threshold $\tau$ can sharply reduce $P$. As expected, $\gamma_1$ increases with the Lipschitz constant $L$ and total noise variance $n(\sigma_1^2 + \sigma_2^2)$, and decreases with the sensitivity $\mathrm{MinDer}_{\mu_1}$. The core technical step is to relate the probability $P$ to the expectation $\mathsf{Err}_{avg}$, using a concentration bound under the independence assumption (Assumption 3.1), via McDiarmid's inequality (Kontorovich, 2014).

**Illustrative example: linear ability profiles.** Consider a random job with $m$ tasks, each requiring $k$ randomly chosen skills from a pool of $n$. Let all aggregation functions $h, g, f$ be averages. In the balanced case, the job error simplifies to: $\mathsf{Err}(\zeta) = \frac{1}{2n} \sum_{j=1}^{n} (\zeta_{j1} + \zeta_{j2})$, with $L = \frac{1}{2n}$. Suppose the ability profile is linear with noise: $\alpha_\ell(s) = 1 - (1 - a_\ell)s + \varepsilon(s)$, where $\varepsilon(s) \sim \min\{1 - (1 - a_\ell)s, (1 - a_\ell)s\} \cdot \mathrm{Unif}[-\sigma, \sigma]$, and assume $s_{j\ell} \sim \mathrm{Unif}[0, 1]$. Then the expected subskill difficulty is $0.5$ and $\mathrm{MinDer}_{\mu_1}(\sigma_1, \mu_2, \sigma_2) = \frac{1}{2n} \sum_j s_{j1} \approx 0.25$. Thus, $\gamma_1 = O(\sigma \sqrt{\ln(1/\theta)/n})$. This implies that elite workers (small $\sigma$) or large jobs (large $n$) experience sharper transitions in job success with ability.

Figure 1 shows this empirically. For $\sigma = 0.1$, increasing $a_1$ by just 4.3% (from 0.492 to 0.513) raises $P$ from 0.2 to 0.8. As $\sigma$ decreases, the transition sharpens, validating our theoretical prediction. Figure 1(c) shows that for jobs

with $P \geq 0.5$, either increasing $a_1$ or reducing $\sigma$ effectively improves success.

**Generalization.** In Section C.1, we prove a generalized form of Theorem 3.2 that accommodates arbitrary noise models $\varepsilon(s)$, leveraging the notion of a *subgaussian constant* to quantify the dispersion of $\varepsilon(s)$. In Section C.2, we further extend the analysis to non-linear aggregation rules (e.g., max) and alternative ability profiles (e.g., constant, polynomial). The resulting transition width $\gamma_1$ varies from $O(1/n)$ to $O(1)$, depending on the functional form and the underlying distributional assumptions.

## 3.3. Merging workers to improve job success

The phase transition result (Theorem 3.2) shows that small increases in ability parameters can sharply increase the success probability $P$. We now apply this insight to demonstrate how *merging* two workers with complementary skills can result in a significant performance gain, especially relevant in settings combining humans and GenAI tools.

Suppose Worker 1 ($W_1$) has stronger decision-level ability, while Worker 2 ($W_2$) excels in action-level execution. Let the decision-level profiles be denoted $\alpha_1^{(\ell)} \sim (\mu_1^{(\ell)}, \sigma_1^{(\ell)})$, and action-level profiles $\alpha_2^{(\ell)} \sim (\mu_2^{(\ell)}, \sigma_2^{(\ell)})$ for $\ell \in \{1, 2\}$. Assume $\mu_1^{(1)} > \mu_1^{(2)}$ and $\mu_2^{(1)} < \mu_2^{(2)}$, i.e., $W_1$ is stronger in decision skills, and $W_2$ in action skills.

We define a *merged worker* $W_{12}$ that uses the decision-level ability of $W_1$ and the action-level ability of $W_2$: $\alpha_1^{(12)} := \alpha_1^{(1)}$, $\alpha_2^{(12)} := \alpha_2^{(2)}$. Let $P_{12}$ denote the success probability of $W_{12}$, and $P_2$ that of $W_2$. We now give conditions under which the merged worker has substantially higher success probability than either of the original workers.

**Theorem 3.3 (Success gain from merging complementary workers).** *Fix the job instance. Let $\theta \in (0, 0.5)$ be a confidence level, and define:*
$$\gamma_1^{(1)} := \frac{L \cdot \sqrt{n\left((\sigma_1^{(1)})^2 + (\sigma_2^{(2)})^2\right) \cdot \ln(1/\theta)}}{\mathrm{MinDer}_{\mu_1}(\sigma_1^{(1)}, \mu_2^{(2)}, \sigma_2^{(2)})}, \text{ and } \gamma_1^{(2)} :=$$
$$\frac{L \cdot \sqrt{n\left((\sigma_1^{(2)})^2 + (\sigma_2^{(2)})^2\right) \cdot \ln(1/\theta)}}{\mathrm{MinDer}_{\mu_1}(\sigma_1^{(2)}, \mu_2^{(2)}, \sigma_2^{(2)})}. \text{ If}$$
$$\mathsf{Err}_{avg}(\mu_1^{(1)} - \gamma_1^{(1)}, \sigma_1^{(1)}, \mu_2^{(2)}, \sigma_2^{(2)}) \leq \tau \leq$$
$$\mathsf{Err}_{avg}(\mu_1^{(2)} + \gamma_1^{(2)}, \sigma_1^{(2)}, \mu_2^{(2)}, \sigma_2^{(2)}),$$

*then under Assumption 3.1, we have: $P_{12} - P_2 \geq 1 - 2\theta$.*

**Gain from merging complementary workers.** If the average error function $\mathsf{Err}_{avg}$ is fully determined by the ability parameters, and $\mathsf{Err}_{avg}(\mu_1^{(1)} - \gamma_1^{(1)}, \sigma_1^{(1)}, \mu_2^{(2)}, \sigma_2^{(2)}) = \tau = \mathsf{Err}_{avg}(\mu_1^{(2)} + \gamma_1^{(2)}, \sigma_1^{(2)}, \mu_2^{(2)}, \sigma_2^{(2)})$, then it follows that $\mu_1^{(1)} = \mu_1^{(2)} + \gamma_1^{(1)} + \gamma_1^{(2)}$. This implies that if $W_1$'s

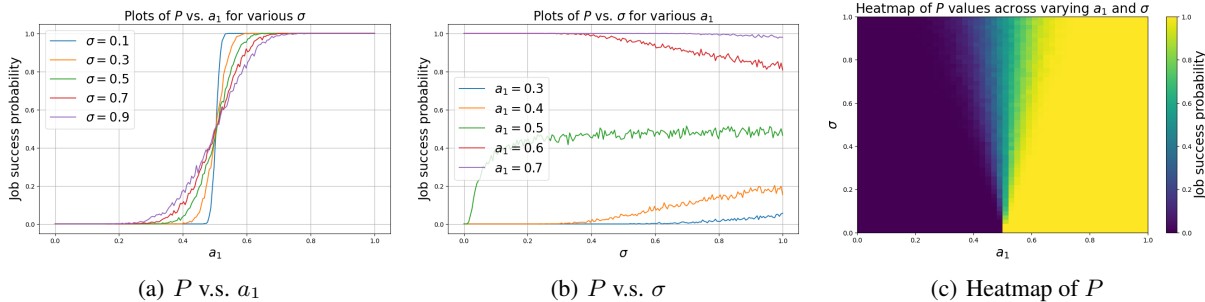

(a) $P$ v.s. $a_1$      (b) $P$ v.s. $\sigma$      (c) Heatmap of $P$

*Figure 1.* Plots illustrating the relationship between the success probability $P(\alpha_1, \alpha_2, h, g, f, \tau)$ and the parameters $a_1, \sigma$ for the linear ability example of Theorem 3.2 with default settings of $(n, m, \tau, a_2) = (20, 20, 0.25, 0.4)$ and subskill numbers $s_{j\ell} \sim \text{Unif}[0, 1]$.

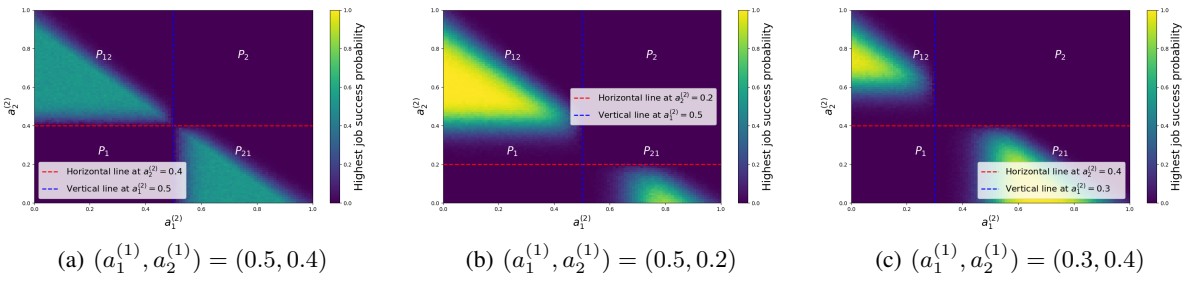

(a) $(a_1^{(1)}, a_2^{(1)}) = (0.5, 0.4)$      (b) $(a_1^{(1)}, a_2^{(1)}) = (0.5, 0.2)$      (c) $(a_1^{(1)}, a_2^{(1)}) = (0.3, 0.4)$

*Figure 2.* Heatmaps of the probability gain $\Delta = \max\{P_1, P_2, P_{12}, P_{21}\} - \max\{P_1, P_2\}$ by merging two workers for different ranges of $(a_1^{(2)}, a_2^{(2)})$ for the linear ability example of Theorem 3.3 with default settings of $(n, m, \sigma, \tau) = (20, 20, 0.5, 0.25)$. The region enclosed by the dotted lines in each heatmap indicates where the corresponding job success probability is the highest among the four. For instance, in Figure 2(a) with $(a_1^{(1)}, a_2^{(1)}) = (0.5, 0.4)$, we observe that when $a_1^{(2)} + a_2^{(2)} < 0.9$ and $a_2^{(2)} > 0.43$, $P_{12}$ is significantly larger than both $P_1$ and $P_2$ by an amount of 0.6. Similarly, when $a_1^{(2)} + a_2^{(2)} < 0.9$ and $a_1^{(2)} > 0.52$, $P_{21}$ is significantly larger than both $P_1$ and $P_2$. We note that the rapid color shifts in the heatmaps reflect an abrupt change in $\Delta$, indicative of a phase transition phenomenon in $P$.

decision-level ability exceeds $W_2$'s by this margin, then their combination $W_{12}$ can substantially outperform $W_2$ alone in job success probability.

**Illustration with linear ability profiles.** Let $\text{Err}(\zeta) = \frac{1}{2n}\sum_{j,\ell}\zeta_{j\ell}$ and assume subskill difficulties $s_{j\ell} \sim \text{Unif}[0,1]$. Let each worker $\ell \in \{1,2\}$ have a linear ability function $\alpha_\ell^{(i)}(s) = 1 - (1 - a_\ell^{(i)})s + \varepsilon(s)$, where $\varepsilon(s) \sim \min\{(1-a_\ell^{(i)})s, 1-(1-a_\ell^{(i)})s\} \cdot \text{Unif}[-\sigma,\sigma]$, and assume a common noise level $\sigma$ for both workers. We analyze when merging $W_1$ and $W_2$ leads to a gain over either alone. If $a_\ell^{(2)} \leq a_\ell^{(1)}$, then $W_1$ is optimal. But if $a_1^{(1)} \geq a_1^{(2)}$ and $a_2^{(1)} \leq a_2^{(2)}$, merging ($P_{12}$) leads to a nontrivial gain.

If $a_1^{(1)} \geq a_1^{(2)} + O(\sigma\sqrt{\ln(1/\theta)/n})$ and $a_2^{(1)} \leq a_2^{(2)} - O(\sigma\sqrt{\ln(1/\theta)/n})$, then by Theorem 3.3, $P_{12} - P_\ell \geq 1 - 2\theta$ for $\ell \in \{1,2\}$. The gain grows as $\sigma$ decreases, making the merging criteria easier to satisfy.

Figure 2 illustrates this effect. For example, when $a_1^{(1)} = 0.5 = a_1^{(2)} + 0.1$ and $a_2^{(1)} = 0.4 = a_2^{(2)} - 0.1$, we observe $P_{12} = 1$ while $P_1 = P_2 = 0.4$, yielding a gain of 0.6.

**Implications.** Our analysis informs both job-worker fit and human-AI collaboration strategies. Theorem 3.2 demon-

strates the impact of targeted upskilling, especially for high-ability, low-variance workers. Section D.1 explores the partial derivative landscape to identify when such interventions are most effective.

If $W_1$ represents a human worker and $W_2$ a GenAI system (as motivated in Section 1), Theorem 3.3 shows that even modest GenAI advantages in action-level tasks can lead to meaningful gains in $P_{12}$. As human action-level ability decreases, $P_1$ falls but $P_{12}$ remains stable, widening the gap $P_{12} - P_1$. This mirrors recent empirical findings (Brynjolfsson et al., 2023; Noy & Zhang, 2023) and contributes to the *productivity compression* effect, further analyzed in Section 3.4. Thus, combining GenAI with human decision-making yields a productivity amplification effect rather than a replacement dynamic. Organizations should invest in decision-level skill development and in reducing ability noise through workflows and training.

Finally, our results also highlight the risk of biased evaluations: underestimating $P$ can exclude strong candidates (see Section D.2). Moreover, realizing the gains of merging hinges on accurate evaluations of both human and AI abilities (see also (Somers, 2023)). Section D.2 also quantifies how imperfect evaluations can negate these gains.

## 3.4. Application: Explaining productivity compression

Brynjolfsson et al. (2023) studied the effect of GenAI tools on customer service productivity, measured by resolutions per hour (RPH). They found that AI assistance disproportionately benefited lower-skilled workers, increasing their RPH by up to 36% and narrowing the productivity gap relative to higher-skilled workers. We now show how Theorem 3.3 provides a theoretical explanation for this effect.

Let $W_1$ and $W_2$ be two human workers with the same families of ability profiles. Assume $\mu_2^{(2)} > \mu_2^{(1)}$, indicating that $W_2$ is more skilled than $W_1$ at the action level. Let $W_{\mathrm{AI}}$ be a GenAI tool sharing the same family of ability profiles as the human workers. For $\ell \in \{1, 2\}$, let $P_\ell$ denote the job success probability of $W_\ell$ before merging with $W_{\mathrm{AI}}$, and let $P'_\ell$ be the corresponding probability after merging. Assuming the job competition time is stable, note that the job success probability $P$ is proportional to the productivity measure RPH (resolutions per hour). Hence, $|P_2 - P_1|$ and $|P'_2 - P'_1|$ represent the productivity gap between $W_1$ and $W_2$ before and after merging, respectively. We define the *productivity compression* as

$$\mathrm{PC} = |P_2 - P_1| - |P'_2 - P'_1|,$$

which measures how much the productivity gap is reduced by merging. A larger $\mathrm{PC}$ indicates that AI assistance more effectively narrows the gap. As a consequence of Theorem 3.3, we obtain the following corollary, deriving conditions on the worker parameters to lower-bound $\mathrm{PC}$.

**Corollary 3.4 (Productivity compression).** *Fix the job instance. Suppose both human workers have the same decision-level abilities:*

$$\mu_1^{(1)} = \mu_1^{(2)} = \mu_1^\star > \mu_1^{(\mathrm{AI})}, \quad \sigma_1^{(1)} = \sigma_1^{(2)} = \sigma_1^{(\mathrm{AI})} = \sigma_1^\star.$$

*Let $\theta \in (0, 0.5)$ be a confidence level, and for each $\ell \in \{1, 2, \mathrm{AI}\}$, define $\gamma_2^{(\ell)} := \frac{L \cdot \sqrt{n\left((\sigma_1^\star)^2 + (\sigma_2^{(\ell)})^2\right) \cdot \ln(1/\theta)}}{\mathrm{MinDer}_{\mu_2}(\sigma_2^{(\ell)}, \mu_1^\star, \sigma_1^\star)}$. If*

$$\max\left\{ \mathrm{Err}_{avg}(\mu_1^\star, \sigma_1^\star, \mu_2^{(\mathrm{AI})} - \gamma_2^{(\mathrm{AI})}, \sigma_2^{(\mathrm{AI})}), \right.$$
$$\left. \mathrm{Err}_{avg}(\mu_1^\star, \sigma_1^\star, \mu_2^{(2)} - \gamma_2^{(2)}, \sigma_2^{(2)}) \right\} \leq \tau \leq$$
$$\mathrm{Err}_{avg}(\mu_1^\star, \sigma_1^\star, \mu_2^{(1)} + \gamma_2^{(1)}, \sigma_2^{(1)}),$$

*then under Assumption 3.1, we have:* $\mathrm{PC} \geq 1 - 2\theta$.

This result implies that if the AI assistant outperforms the lower-skilled worker by at least $\gamma_2^{(1)} + \gamma_2^{(\mathrm{AI})}$, the productivity gap can shrink significantly. To our knowledge, this is one of the first formal models explaining the productivity compression effect under realistic assumptions.

In Section C.4, we provide the proof of Corollary 3.4 and further extend this analysis to the case where the GenAI tool uses a different ability profile family, confirming that our framework generalizes beyond identical distributions.

## 4. Empirical results

We demonstrate the usability of our framework using real-world data and validate our theoretical findings in both noise-dependent settings and scenarios involving the merging of workers with distinct ability profiles. Key results are summarized below, with full implementation details provided in Section E. We further validate the robustness of our findings across alternative modeling choices in Section E.4.

### 4.1. Data, subskills, abilities, and parameters

We derive job and worker data from O*NET and Big-bench Lite. To bridge missing parameters, we introduce a general subskill division method. As a running example, consider the job of Computer Programmers.

*Deriving job data.* O*NET states that the Computer Programmer job as requires $n = 18$ skills and $m = 17$ tasks, and provides their descriptions. It also gives proficiency levels for each skill, represented by $s = (.41, .43, .45, .45, .45, .46, .46, .46, .46, .48, .5, .5, .52, .54, .55, .55, .57, .7)$, where $s_j \in [0, 1]$ denotes the skill's criticality for the job. O*NET also provides task and skill importance scores, which inform the choice of $g$ and $f$.

*Deriving workers' abilities.* We begin by considering lab evaluations from Big-bench Lite (bench authors, 2023) for both human workers and GenAI tools (specifically PaLM (Chowdhery et al., 2023)). For example, we model the ability profiles of a human worker ($W_1$) and a GenAI tool ($W_2$) as $\alpha^{(1)}(s) = \mathrm{TrunN}(1 - 0.78s + 0.22, 0.013; 0, 1)$, and $\alpha^{(2)}(s) = \mathrm{TrunN}(1 - 0.92s + 0.08, 0.029; 0, 1)$.

*An approach for subskill division.* Subskill division becomes essential for analyzing worker performance. We first use GPT-4o to determine the decision-level degree for each skill, given by $\lambda = (0, 0, 1, 1, 1, .6, .7, .4, .4, 0, .3, 1, 1, .6, .7, .6, 0, .4)$. Using a skill proficiency $s_j$ and its decision-level degree $\lambda_j$, we compute subskill numbers as $s_{j1} = \lambda_j s_j$, $s_{j2} = (1 - \lambda_j)s_j$. These values are listed in Eq. (11). This formulation ensures that subskill numbers $s_{j1}$ and $s_{j2}$ are linear functions of $s_j$ and $\lambda_j$, maintaining the property that $s_{j1} + s_{j2} = s_j$. Further examples are in Section E.3.

We decompose skill ability profiles $\alpha$ into subskill ability profiles $\alpha_1$ and $\alpha_2$. For $\alpha(s) \sim \mathrm{TrunN}(1 - (1 - a)s, \sigma^2; 0, 1)$ with decision-level degree $\lambda \in [0, 1]$, we set $\alpha_1(s) = \alpha_2(s) = \mathrm{TrunN}(1 - (1 - a)s, \sigma^2/2; 0, 1)$, so that the distribution of $\zeta_{j1} + \zeta_{j2}$ approximates first drawing $X \sim \alpha(s_j)$ and then outputting $1 - X$. Thus, skill profiles can be (approximately) reconstructed by setting the skill success probability function as $h(\zeta_1, \zeta_2) = \zeta_1 + \zeta_2$. Thus, we obtain $\alpha_\ell^{(1)}(s) = \mathrm{TrunN}(1 - 0.78s, 0.0065; 0, 1)$, $\alpha_\ell^{(2)}(s) = \mathrm{TrunN}(1 - 0.92s, 0.0145; 0, 1)$.

*Constructing the task-skill dependency.* Using task and skill descriptions from O*NET, we employ GPT-4o to generate task-skill dependencies $T_i \subseteq [n]$ for each task $i \in [m]$. Details are provided in Section E.3.

*Choice of error functions and threshold.* We set the skill error function as $h(\zeta_1, \zeta_2) = \zeta_1 + \zeta_2$ to ensure consistency with the skill ability function $\alpha$ derived from Big-bench Lite. Task and job error functions, $g$ and $f$, are weighted averages based on the importance of skills and tasks from O*NET, resulting in $\text{Err}(\zeta) = \sum_{j \in [n]} w_j(\zeta_{j1} + \zeta_{j2})$. (See Eq.(13) for details.) We set the threshold $\tau = 0.45$, representing a medium job requirement.

*Summary.* In this manner, all necessary job and worker attributes can be extracted from sources such as O*NET and Big-bench Lite, with GPT-4o (or similar models) assisting in estimating skill proficiencies, decision-level intensities, and task-skill mappings. This subskill decomposition method is generic and can be applied to other job and worker datasets, making it practical across diverse domains.

We note that O*NET and Big-bench Lite offer complementary but biased views of work. O*NET emphasizes tasks involving judgment, creativity, and interpersonal skills, potentially under-representing emerging digital or computational activities. Conversely, Big-bench Lite focuses on structured, rule-based problems where GenAI systems tend to excel. Empirical insights should therefore be interpreted in light of these distributions, as each dataset highlights different aspects of human–AI complementarity.

### 4.2. Evaluating worker-job fit with dependent abilities

Theorem 3.2 assumes independent subskill abilities (Assumption 3.1), but this may not hold in practice. For instance, a worker's current state—such as fatigue or motivation—can influence their abilities (J. et al., 1976), creating dependencies between subskill error rates $\zeta_{j\ell}$. This raises the question: Under such dependencies, can a slight increase in ability still lead to a dramatic nonlinear rise in success probability?

*Choice of parameters.* We set $\alpha_1(s) = \text{TrunN}(1 - (1 - a)s, 0.0065; 0, 1)$ and $\alpha_2(s) = \text{TrunN}(1 - 0.78s, 0.0065; 0, 1)$ to model a human worker, where the parameter $a$ controls the decision-level ability. For $s \in [0, 1]$, let $F_s$ denote the cumulative density function of $\alpha_1(s)$. To introduce dependency between subskills, we assume that the worker has a random status $\beta \sim \text{Unif}[0, 1]$. For each subskill, $\zeta_{j\ell} \sim 1 - \alpha_\ell(s_{j\ell})$ with probability $1 - p$ and $\zeta_{j\ell} = 1 - F_{s_{j\ell}}^{-1}(\beta)$ with probability $p$. As $p$ increases, the dependency between the $\zeta_{j\ell}$s strengthens. Specifically, when $p = 0$, all $\zeta_{j\ell}$s are independent. Conversely, when $p = 1$, all $\zeta_{j\ell}$s are fully determined by the worker's status $\beta$, making them highly correlated.

*Analysis.* We plot the job success probability $P$ in Figure 3 as the ability parameter $a$ and dependency parameter $p$ vary. Figure 3(a) shows that phase transitions in $P$ persist even when subskills are dependent ($p > 0$), although the transition window narrows as $p$ decreases. For example, when $p = 0$, increasing $a$ by 0.27 (from 0.07 to 0.34) raises $P$ from 0.2 to 0.8, whereas for $p = 0.4$, a greater increase in $a$ (0.44) is needed. Figure 3(b) shows that for fixed $a$, $P$ increases monotonically with $p$ when $P < 0.5$ and decreases monotonically when $P > 0.5$, similar to the trend in $P$ v.s. $\sigma$ (Figure 1(b)). This is because the variance of $\text{Err}(\zeta)$ increases with both $p$ and $\sigma$. These results show that workers with loosely coupled subskills (low $p$) experience sharper gains in $P$ from ability improvements, underscoring the value of reducing skill interdependencies.

### 4.3. Merging two workers with distinct ability profiles

We empirically examine the utility of merging two workers ($W_1$ and $W_2$). Theorem 3.3 assumes identical ability profile families, ensuring that $W_1$ consistently outperforms $W_2$ across all decision-level (action-level) subskills, or vice versa. In practice, however, this may not hold—e.g., a GenAI tool may surpass a human in some action-level subskills but not others. This raises the question: Does the sharp increase in job success probability from merging persist when workers have ability profiles from different families?

*Choice of parameters.* We set the subskill ability profiles of $W_1$ to be linear: $\alpha_1^{(1)}(s) = \alpha_2^{(1)}(s) = \text{TrunN}(1 - 0.78s, 0.0065; 0, 1)$, representing a human worker. For the second worker ($W_2$), we define $\alpha_1^{(2)} = \text{TrunN}(1 - (1 - a)s, 0.0145; 0, 1)$ and $\alpha_2^{(2)} = \text{TrunN}(c, 0.0145; 0, 1)$. This models a GenAI tool that excels at easier decision-level subskills but degrades with difficulty, while maintaining strong and uniform action-level abilities.

We analyze which decision- and action-level subskills should be assigned to each worker and quantify the resulting gain in job success probability.

If the average of $\alpha_1^{(1)}(s)$ exceeds that of $\alpha_1^{(2)}(s)$ (i.e., $a < 0.22$), all decision-level subskills are assigned to $W_1$; otherwise ($a \geq 0.22$), to $W_2$. Because the two workers' action-level abilities differ non-monotonically, neither dominates the other across all subskills. This renders the uniform merging strategy from Section 3.3 sub-optimal. Instead, we select the action-level subskill provider based on difficulty: the average of $\alpha_2^{(1)}(s)$ is $1 - 0.78s$, while for $\alpha_2^{(2)}(s)$ it is constant at $c$. Thus, for $s_{j2} \leq \frac{1-c}{0.78}$, $W_1$ has higher expected ability and is chosen; otherwise, $W_2$ is selected. This creates a merged worker $W_{\text{merge}}$ whose decision-level ability is linear and action-level ability is piecewise linear with a breakpoint at $s_{j2} = \frac{1-c}{0.78}$. Let $P_{\text{merge}}$ denote the job success probability of this merged worker.

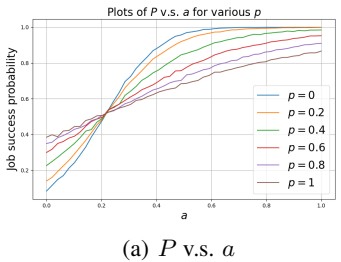
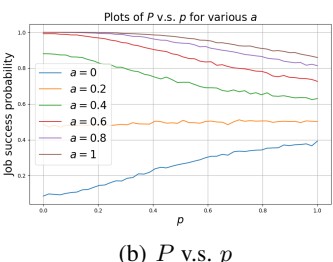
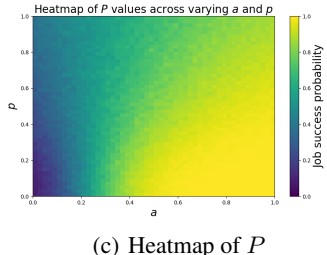

(a) $P$ v.s. $a$        (b) $P$ v.s. $p$        (c) Heatmap of $P$

*Figure 3.* Plots illustrating the relationship between the success probability $P(\alpha_1, \alpha_2, h, g, f, \tau)$ and the ability parameter $a$ and dependency parameter $p$ for the Computer Programmer example with default settings of $(\sigma, \tau) = (0.08, 0.45)$.

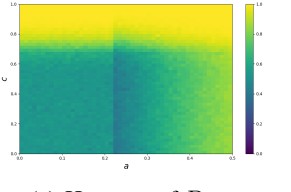
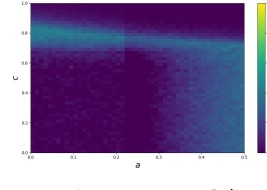

(a) Heatmap of $P_{merge}$      (b) Heatmap of $\Delta$

*Figure 4.* Heatmaps of the merged job success probability $P_{\text{merge}}$ and the corresponding probability gain $\Delta = P_{\text{merge}} - \max\{P_1, P_2\}$, shown across different values of the ability parameters $(a, c)$ for the Computer Programmers example with default threshold $\tau = 0.45$. Rapid color transitions reflect persistent phase shifts in both $P_{\text{merge}}$ and $\Delta$, even when worker profiles differ. Compared to Figure 2, the narrower bright region in Figure 4(b) suggests that merging distinct profiles yields more gradual improvements than merging identical ones.

*Analysis.* Figure 4.2 plots the heatmaps of job success probability $P_{merge}$ and probability gains $\Delta = P_{merge} - \max\{P_1, P_2\}$ as ability parameters $a$ and $c$ vary. When $a \leq 0.22$ (i.e., $W_2$ has lower decision-level ability than $W_1$) and $c \in [0.78, 0.82]$, we observe $P_{merge} = 1$ while $P_1, P_2 \leq 0.6$, indicating a probability gain of at least $P_{merge} - P_\ell \geq 0.4$. This occurs because $c$ first reaches 0.78, triggering a sharp increase in $P_{merge}$ as predicted by Theorem 3.2, and later reaches 0.82, aligning $P_2$ with the trend in Figure 2. The range of $c$ is narrower than that of $\alpha_2^{(2)}$ in Figure 2 since increasing $c$ results in a smooth transition in action-level subskills from $W_1$ to $W_2$. Conversely, when $\alpha_2^{(2)}$ surpasses $\alpha_2^{(1)}$, all action-level subskills shift abruptly, causing a more sudden transition. These findings confirm that the nonlinear probability gain from merging persists even when workers specialize in different action-level subskills, affirming our hypothesis.

## 5. Conclusions, limitations, and future work

This work examines the evolving impact of GenAI tools in the workforce by introducing a mathematical framework to assess job success probability in worker-job configurations. By decomposing skills into decision-level and action-level subskills, the framework enables fine-grained analysis and

offers insights into effective human–AI collaboration. Our theoretical results identify conditions under which job success probability changes sharply with worker ability, and show that merging workers with complementary subskills can substantially enhance performance, reinforcing the view that GenAI tools augment, rather than replace, human expertise. This includes explaining the phenomenon of *productivity compression*, where GenAI assistance disproportionately benefits lower-skilled workers, narrowing performance gaps, consistent with empirical findings from recent field studies.

We demonstrate how the framework integrates with real-world datasets such as O*NET and Big-bench Lite, highlighting its practical relevance. Empirical results validate theoretical insights, even under relaxed assumptions.

Our analysis focuses primarily on job success probability. In practice, performance also depends on factors such as efficiency, time, and cost. Incorporating these dimensions would yield a more comprehensive view of worker-job fit and inform workforce optimization strategies.

Moreover, the datasets used may not fully capture the complexity of skill attribution in dynamic work settings. O*NET reflects static, survey-based assessments, while LLM-based estimates from Big-bench may embed modeling biases. Incorporating empirical benchmarks (e.g., HumanEval for coding, customer support transcripts) could strengthen the framework's empirical grounding.

Our model underscores the importance of improving evaluation mechanisms to better reflect the strengths and limitations of human and AI capabilities. More broadly, this work contributes to the growing literature on AI and work by offering a quantitative lens to study the interplay between human expertise and GenAI systems. As AI continues to reshape labor markets, balancing human skill and automation remains a critical challenge.

This paper offers a step toward quantifying that balance; further research is needed to refine models, incorporate behavioral studies, and promote equitable and effective human–AI collaboration in an evolving workplace.

## Impact statement

This paper introduces a framework for assessing worker-job fit for both human and AI workers, aiming to advance workforce optimization in an era of rapid technological change. Our insights and methodologies contribute to more effective allocation of human and AI resources, improving job success probability and facilitating productive human-AI collaboration.

The societal implications are twofold. On one hand, the framework empowers organizations to make data-driven workforce decisions, enhancing productivity and job satisfaction. On the other, it highlights challenges such as biases in ability evaluation and the evolving role of GenAI in labor markets, underscoring the need for careful consideration to prevent exclusion or unfair treatment of workers.

While our work may influence hiring practices and perceptions of human-AI collaboration, these outcomes should be interpreted within the broader goal of equitable and efficient workforce optimization. We do not identify any immediate ethical concerns beyond these considerations.

## Acknowledgments

This work was funded by NSF Awards IIS-2045951 and CCF-2112665, and in part by grants from Tata Sons Private Limited, Tata Consultancy Services Limited, Titan, and New Cornerstone Science Foundation.

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

## A. Additional related work

Workforce optimization seeks to align employee skills with organizational objectives to improve productivity, efficiency, and satisfaction (Sinclair & for Employment Studies, 2004; wik; Services, 2024; Naveh et al., 2007). Although this area has been extensively studied empirically, theoretical models that systematically evaluate worker-job fit remain limited.

The emergence of GenAI tools has reignited debates around automation and its impact on skilled labor (Dillion et al., 2023; Harding et al., 2023; Stade et al., 2024). Recent work demonstrates AI's proficiency in complex tasks—from expert-level reasoning (OpenAI, 2024; Will Knight, 2024) to high performance on domain-specific benchmarks such as LiveBench and MMLU-Pro (White et al., 2024; Wang et al., 2024). These advances underscore the need for principled frameworks to assess human–AI complementarity (Yu et al., 2024; Hendrycks et al., 2021).

Several studies compare human and AI capabilities across domains (Brynjolfsson et al., 2023; Noy & Zhang, 2023; Sharma, 2024; Eloundou et al., 2023). However, existing models often conflate decision-making and execution, overlooking their distinct roles in work. Our framework explicitly separates decision-level and action-level subskills, enabling a more granular analysis of how AI systems complement human abilities.

Research on the labor implications of AI suggests that GenAI tends to augment lower-skilled workers (Acemoglu & Autor, 2011), consistent with our finding that AI enhances action-level subskills while decision-level abilities remain critical. Studies on AI-driven productivity gains (Noy & Zhang, 2023; Lo et al., 2024; Fosso Wamba et al., 2023) offer additional empirical support for our model's predictions. Integrating real-world data to further validate our theoretical insights is an important direction for future work.

Recent work by Acemoglu (2025) presents a macroeconomic model that analyzes the effects of AI on productivity, wages, and inequality via equilibrium-based task allocation between labor and capital. While developed independently, their assumptions—such as task decomposition, differential AI performance across task types, and heterogeneity in worker productivity—resonate with our framework's decomposition of skills and modeling of worker ability. The key distinction lies in scope: their model addresses aggregate, market-level outcomes, whereas ours focuses on job-level success and collaboration between individual workers.

## B. Properties of ability profiles

This section discusses the properties of several ability profiles.

### B.1. Monotonicity, variability, and visualization for ability profiles

Below, we formalize three ability profiles: constant, linear, and polynomial with additive uniform noise.

- **Constant profile.** We consider a *constant profile* $\alpha \equiv c + \varepsilon$ for some $c \in [0, 1]$ and $\varepsilon$ distributed according to a certain noise distribution, e.g., $\varepsilon \sim \min\{c, 1-c\} \cdot \mathrm{Unif}[-\sigma, \sigma]$ for some $\sigma \in [0, 1]$. When $\sigma = 0$, $\alpha \equiv c$ is a constant function. This model is useful for skills where ability does not vary with increased experience, such as automated processes handled by GenAI tools, where the output remains consistent regardless of operational duration. The scale of two parameters $c$ and $\sigma$ determines the performance of $\alpha$. Parameter $c$ determines the average ability of constant profiles, where $c = 0$ represents the worst ability and $c = 1$ represents the best ability. Moreover, fixing $\sigma$, the ability of a constant profile rises smoothly from the worst to the best as $c$ increases from 0 to 1. Also note that $\mathrm{Var}[\alpha(s)] = \min\{c, 1-c\} \cdot \frac{\sigma^2}{3}$. Thus, parameter $\sigma$ reflects the variability of $\alpha$.

- **Linear profile.** We consider a *linear profile* $\alpha(s) = c - 1 - (1-a)s + \varepsilon(s)$ for some $a, c \in [0, 1]$ and noise $\varepsilon(s) \sim \min\{(1-a)s, 1-(1-a)s\} \cdot \mathrm{Unif}[-\sigma, \sigma]$ for some $\sigma \in [0, 1]$. When $a > 0$, this model is apt for scenarios involving workers whose ability to develop a skill increases linearly with the ease of the skill. Note that for any $s \in [0, 1]$, $E(s) = c - (1-a)s$ is a monotone increasing function of both $c$ and $a$. Thus, the average ability of a linear profile rises smoothly from the worst to the best as $a$ (or $c$) increases from 0 to 1. In this paper, we usually set $c = 1$ such that $E(0) = 1$, representing the highest ability for the easiest subskill.

- **Polynomial profile.** We consider a *polynomial profile* $\alpha(s) = 1 - s^\beta + \varepsilon(s)$, for some parameter $\beta \geq 0$ and noise $\varepsilon(s) \sim \min\{s^\beta, 1-s^\beta\} \cdot \mathrm{Unif}[-\sigma, \sigma]$ for some $\sigma \in [0, 1]$. This function can be used to model how decision-making subskills often improve nonlinearly with the easiness of the skill. The unique parameter, $\beta$, allows us to adjust the

sensitivity of the ability function to changes in $s$, where $\beta = 0$ represents the worst ability $\alpha \equiv \varepsilon(s)$ and $\beta = \infty$ represents the best ability $\alpha \equiv 0$. Moreover, higher values of $\beta$ indicate a more pronounced increase in average ability as $s$ approaches 1.

We plot constant, linear, and polynomial profiles with additive uniform noise in Figure 5. In these models, we know that ability profiles are usually characterized by a parameter measuring the average ability (e.g., $c$ for constant profiles, $a$ for linear profiles, and $\beta$ for polynomial profiles) and a parameter $\sigma$ measuring the variability. A key problem in this paper is quantifying how these parameters affect the performance of workers in a given job. Intuitively, the quality of an ability profile typically improves as the average ability increases or the variability decreases.

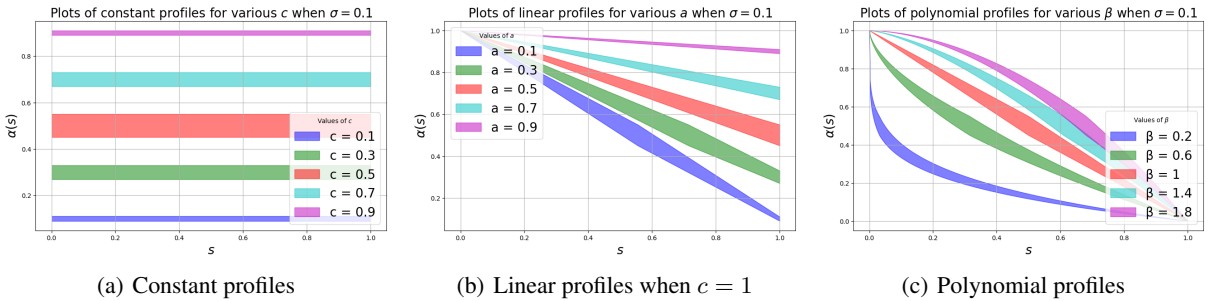

| (a) Constant profiles | (b) Linear profiles when $c = 1$ | (c) Polynomial profiles |

*Figure 5.* Plots by varying the ability parameter for various families of ability profiles with an additive uniform noise when the noise level $\sigma = 0.1$. Observe that the width of the domain $\alpha(s)$ is the largest when the average ability $E(s) = 0.5$ and is the smallest (0) when $E(s) \equiv 0$ or 1. This noise level indicates that the best worker always completes the subskill flawlessly ($\alpha(s) \equiv 1$), the worst worker always fails at the subskill ($\alpha(s) \equiv 0$), while the performance of a medium worker exhibits greater variability.

### B.2. Stochastic dominance for ability profiles

We first define the stochastic dominance property for ability profiles.

**Definition B.1** (**Stochastic dominance for ability profiles**). *Let $\alpha, \alpha'$ be two ability profiles parameterized by ability parameter $\mu, \mu'$, respectively, and the same noise parameter $\sigma$. Suppose $\mu \leq \mu'$. We say $\alpha$ has stochastic dominance over $\alpha'$ if for any $s \in [0, 1]$ and $x \geq 0$:*

$$\Pr_{\zeta \sim \alpha(s)}[\zeta \geq x] \leq \Pr_{\zeta' \sim \alpha'(s)}[\zeta' \geq x].$$

We propose the following proposition that shows that the studied ability profiles have stochastic dominance properties.

**Proposition B.1** (**Stochastic dominance for ability profiles**). *The following hold:*

- **Constant profile** *Let $\alpha \equiv c + \varepsilon$ for some $c \in [0, 1]$ and $\varepsilon \sim \min\{c, 1 - c\} \cdot \mathrm{Unif}[-\sigma, \sigma]$ for some $\sigma \in [0, 1]$. Let $\alpha' \equiv c' + \varepsilon(s)$ for some $c' \in [0, 1]$ with $c \leq c'$ and $\varepsilon(s) \sim \mathrm{Unif}[-\sigma, \sigma]$. Then $\alpha$ has stochastic dominance over $\alpha'$.*

- **Slope parameter for linear profile** *Let $\alpha(s) = 1 - (1 - a)s + \varepsilon(s)$ for some $a \in [0, 1]$ and noise $\varepsilon(s) \sim \min\{(1 - a)s, 1 - (1 - a)s\} \cdot \mathrm{Unif}[-\sigma, \sigma]$. Let $\alpha'(s) = 1 - (1 - a')s + \varepsilon(s)$ for some $a' \in [0, 1]$ with $a \leq a'$ and noise $\varepsilon(s) \sim \mathrm{Unif}[-\min\{(1 - a')s, 1 - (1 - a')s\}\sigma, \min\{(1 - a')s, 1 - (1 - a')s\}\sigma]$. Then $\alpha$ has stochastic dominance over $\alpha'$.*

- **Polynomial profile** *Let $\alpha(s) = 1 - s^{\beta} + \varepsilon(s)$ for some $\beta \in [0, 1]$ and noise $\varepsilon(s) \sim \min\{s^{\beta}, 1 - s^{\beta}\} \cdot \mathrm{Unif}[-\sigma, \sigma]$. Let $\alpha'(s) = 1 - s^{\beta'} + \varepsilon(s)$ for some $\beta' \in [0, 1]$ with $\beta \leq \beta'$ and noise $\varepsilon(s) \sim \mathrm{Unif}[-\min\{s^{\beta'}, 1 - s^{\beta'}\}\sigma, \min\{s^{\beta'}, 1 - s^{\beta'}\}\sigma]$. Then $\alpha$ has stochastic dominance over $\alpha'$.*

The first two items ensure that ability profiles satisfy stochastic dominance with respect to both ability parameters $c$ and $a$.

*Proof of Proposition B.1.* We prove for each family of ability profiles.

**Constant profile.** For any $x \geq 0$, we have

$$\Pr_{\zeta \sim \alpha(s)} [\zeta \geq x] = \min \left\{ 1, \max \left\{ 0, \frac{x - (1 - c) + \min \{c, 1 - c\} \cdot \sigma}{2 \min \{c, 1 - c\} \cdot \sigma} \right\} \right\},$$

Note that when $1 - c \leq 0.5$, we have

$$\frac{x - (1 - c) + \min \{c, 1 - c\} \cdot \sigma}{2 \min \{c, 1 - c\} \cdot \sigma} = -\frac{1}{2\sigma} + 0.5 + \frac{x}{2(1 - c)\sigma},$$

which is increasing with $c$. When $1 - c > 0.5$, we have

$$\frac{x - (1 - c) + \min \{c, 1 - c\} \cdot \sigma}{2 \min \{c, 1 - c\} \cdot \sigma} = \frac{1}{2\sigma} + 0.5 - \frac{1 - x}{2c\sigma},$$

which is increasing with $c$. Overall, $\Pr_{\zeta \sim \alpha(s)} [\zeta \geq x]$ is increasing with $c$. Thus, since $c \leq c'$, we have $\Pr_{\zeta \sim \alpha(s)} [\zeta \geq x] \leq \Pr_{\zeta' \sim \alpha'(s)} [\zeta' \geq x]$.

**Linear profile.** For any $x \in [0, 1]$ and $s \in [0, 1]$, we have

$$\Pr_{\zeta \sim \alpha(s)} [\zeta \geq x] = \min \left\{ 1, \max \left\{ 0, \frac{x - (1 - a)s + \min \{(1 - a)s, 1 - (1 - a)s\} \sigma}{2 \min \{(1 - a)s, 1 - (1 - a)s\} \sigma} \right\} \right\}.$$

Note that when $(1 - a)s \leq 0.5$, we have

$$\frac{x - (1 - a)s + \min \{(1 - a)s, 1 - (1 - a)s\} \sigma}{2 \min \{(1 - a)s, 1 - (1 - a)s\} \sigma} = -\frac{1}{2\sigma} + 0.5 + \frac{x}{2(1 - a)s\sigma},$$

which is increasing with $a$. When $(1 - a)s > 0.5$, we have

$$\frac{x - (1 - a)s + \min \{(1 - a)s, 1 - (1 - a)s\} \sigma}{2 \min \{(1 - a)s, 1 - (1 - a)s\} \sigma} = \frac{1}{2\sigma} + 0.5 - \frac{1 - x}{2(1 - (1 - a)s)\sigma},$$

which is increasing with $a$. Overall, $\Pr_{\zeta \sim \alpha(s)} [\zeta \geq x]$ is increasing with $a$. Since $a \leq a'$, we have $\Pr_{\zeta \sim \alpha(s)} [\zeta \geq x] \leq \Pr_{\zeta' \sim 1 - \alpha'(s)} [\zeta' \geq x]$.

**Polynomial profile.** For any $x \in [0, 1]$ and $s \in [0, 1]$, we have

$$\Pr_{\zeta \sim \alpha(s)} [\zeta \geq x] = \min \left\{ 1, \max \left\{ 0, \frac{x - s^\beta + \min \{s^\beta, 1 - s^\beta\} \sigma}{2 \min \{s^\beta, 1 - s^\beta\} \sigma} \right\} \right\}.$$

Note that when $s^\beta \leq 0.5$, we have

$$\frac{x - s^\beta + \min \{s^\beta, 1 - s^\beta\} \sigma}{2 \min \{s^\beta, 1 - s^\beta\} \sigma} = -\frac{1}{2\sigma} + 0.5 + \frac{x}{2s^\beta \sigma},$$

which is increasing with $\beta$. When $s^\beta > 0.5$, we have

$$\frac{x - s^\beta + \min \{s^\beta, 1 - s^\beta\} \sigma}{2 \min \{s^\beta, 1 - s^\beta\} \sigma} = \frac{1}{2\sigma} + 0.5 - \frac{1 - x}{2(1 - s^\beta)\sigma},$$

which is increasing with $\beta$. Overall, $\Pr_{\zeta \sim \alpha(s)} [\zeta \geq x]$ is increasing with $\beta$. Since $\beta \leq \beta'$, we have $\Pr_{\zeta \sim \alpha(s)} [\zeta \geq x] \leq \Pr_{\zeta' \sim \alpha'(s)} [\zeta' \geq x]$.

Thus, we complete the proof of Proposition B.1. $\qquad \square$

## C. Proofs of results in Section 3 and extensions

In this section, we provide the omitted proofs of the results in Section 3 and demonstrate how to extend them to accommodate general and dependent noise models. We further extend the analysis of linear ability profiles from Section 3.2 to alternative choices of job error functions and ability profile families (see Section C.2).

## C.1. Proof of Theorem 3.2: Phase transition in success probability

We prove a generalized version of Theorem 3.2 that accommodates arbitrary noise models $\varepsilon(s)$, extending beyond the uniform and truncated normal distributions introduced in Section 2.

We first need the following notion that captures the dispersion degree of ability profiles $(\alpha_1, \alpha_2)$.

**Definition C.1** (**Subgaussian constant and maximum dispersion**). *Let $\alpha_\ell$ be parameterized by $\mu_\ell, \sigma_\ell \geq 0$. For each subskill $s_{j\ell}$, define the smallest constant $\mathrm{sg}_{j\ell}(\mu_\ell, \sigma_\ell)$ such that for all $\beta \in \mathbb{R}$,*

$$\mathbb{E}_{X \sim \alpha_\ell(s_{j\ell})} \left[ e^{\beta(X - \mathbb{E}[X])} \right] \leq \exp\left(\frac{\mathrm{sg}_{j\ell}(\mu_\ell, \sigma_\ell)^2 \beta^2}{2}\right).$$

*We define the subgaussian constant as:*

$$\mathrm{sg}(\mu_1, \sigma_1, \mu_2, \sigma_2) := \sum_{j \in [n], \ell \in \{1,2\}} \mathrm{sg}_{j\ell}(\mu_\ell, \sigma_\ell)^2.$$

*Given $\sigma_1, \mu_2, \sigma_2$, the maximum dispersion over $\mu_1$ is defined as:*

$$\mathrm{MaxDisp}_{\mu_1}(\sigma_1, \mu_2, \sigma_2) := \sup_{\mu_1 \geq 0} \mathrm{sg}(\mu_1, \sigma_1, \mu_2, \sigma_2).$$

$\mathrm{MaxDisp}_{\mu_2}(\sigma_2, \mu_1, \sigma_1)$ *is defined similarly.*

Intuitively, the subgaussian constant $\mathrm{sg}_{j\ell}$ quantifies the variability of a subskill's ability distribution around its mean (van Handel, 2014; Vershynin, 2018). The maximum dispersion $\mathrm{MaxDisp}$ captures the cumulative uncertainty across all subskills by aggregating the subgaussian parameters. For example, under uniform noise $\varepsilon(s)$, we have $\mathrm{sg}_{j\ell} \leq \sigma_\ell^2/4$, yielding $\mathrm{MaxDisp}_{\mu_1}(\sigma_1, \mu_2, \sigma_2) \leq n(\sigma_1^2 + \sigma_2^2)/4$. Under truncated normal noise, $\mathrm{sg}_{j\ell} \leq \sigma_\ell^2$, giving $\mathrm{MaxDisp}_{\mu_1}(\sigma_1, \mu_2, \sigma_2) \leq n(\sigma_1^2 + \sigma_2^2)$. As the noise parameters $\sigma_\ell$ increase, $\mathrm{MaxDisp}$ also increases, reflecting greater dispersion in ability. In the deterministic case where $\sigma_1 = \sigma_2 = 0$, we have $\mathrm{MaxDisp} = 0$, indicating no uncertainty in abilities.

We propose the following generalized version of Theorem 3.2, where the term $n(\sigma_1^2 + \sigma_2^2)$ in $\gamma_1$ (designed for both uniform and truncated normal noises) is replaced by the more general quantity $\mathrm{MaxDisp}_{\mu_1}(\sigma_1, \mu_2, \sigma_2)$.

**Theorem C.1** (**Extension of Theorem 3.2 to general noise models**). *Fix the job instance, action-level ability $\mu_2$, and noise levels $\sigma_1, \sigma_2$. Let $\mu_1^c$ be the unique value such that the expected job error equals the success threshold:*

$$\mathsf{Err}_{avg}(\mu_1^c, \sigma_1, \mu_2, \sigma_2) = \tau.$$

*Let $\theta \in (0, 0.5)$ be a confidence level, and define the transition width: $\gamma_1 := \frac{L\sqrt{\mathrm{MaxDisp}_{\mu_1}(\sigma_1, \mu_2, \sigma_2) \cdot \ln(1/\theta)}}{\mathrm{MinDer}_{\mu_1}(\sigma_1, \mu_2, \sigma_2)}$, where $L$ is the Lipschitz constant of the job error function. Then the job success probability satisfies:*

$$P \leq \theta \ \text{ if } \ \mu_1 \leq \mu_1^c - \gamma_1 \text{ and } P \geq 1 - \theta \ \text{ if } \ \mu_1 \geq \mu_1^c + \gamma_1.$$

For preparation, we first introduce the following variant of McDiarmid's inequality. Given a random variable $X$ on $\mathbb{R}$, we define $\|X\|_{\psi_2}$ to be the smallest number $a \geq 0$ such that for any $\beta \in \mathbb{R}$, $\mathbb{E}\left[e^{\beta X}\right] \leq e^{\beta^2 a^2/2}$.

**Theorem C.2** (**Refinement of Theorem 1 in** (Kontorovich, 2014)). *Let $G : \mathbb{R}^T \to \mathbb{R}$ be a 1-lipschitz function. Suppose $X_1, \ldots, X_T$ are independent random variables. Then we have for any $t > 0$,*

$$\Pr_{X_1, \ldots, X_T} [G(X_1, \ldots, X_T) \geq \mathbb{E}[G(X_1, \ldots, X_T)] + t] \leq e^{-\frac{2t^2}{\sum_{j \in [T]} \|X_j - \mathbb{E}[X_j]\|_{\psi_2}^2}},$$

*and*

$$\Pr_{X_1, \ldots, X_T} [G(X_1, \ldots, X_T) \leq \mathbb{E}[G(X_1, \ldots, X_T)] - t] \leq e^{-\frac{2t^2}{\sum_{j \in [T]} \|X_j - \mathbb{E}[X_j]\|_{\psi_2}^2}}.$$

The theorem provides a concentration bound for the function value of $G$ when its input variables are independent subgaussian. Now we are ready to prove Theorem C.1.

*Proof of Theorem C.1.* Recall that Err is a function of $2n$ realized subskill abilities $\zeta_{j\ell}$. By Assumption 3.1, $\zeta_{j\ell}$s are independent random variables. Moreover, by Definition C.1, we know that

$$\mathrm{sg}_j(\mu_\ell, \sigma_\ell) = \| \zeta_{j\ell} - \mathbb{E}_{\zeta \sim 1-\alpha_\ell(s_{j\ell})}[\zeta] \|_{\psi_2}^2,$$

and hence,

$$\mathrm{sg}(\mu_1, \sigma_1, \mu_2, \sigma_2) = \sum_{j\in[n], \ell\in\{1,2\}} \mathrm{sg}_j(\mu_\ell, \sigma_\ell) = \sum_{j\in[n], \ell\in\{1,2\}} \| \zeta_{j\ell} - \mathbb{E}_{\zeta \sim 1-\alpha_\ell(s_{j\ell})}[\zeta] \|_{\psi_2}^2.$$

Since Err is $L$-lipschitz, function $\frac{1}{L} \cdot$ Err is 1-lipschitz. Also, recall that $\mathrm{Err}_{\mathrm{avg}}(\mu_1, \sigma_1, \mu_2, \sigma_2) := \mathbb{E}_{\zeta_{j\ell} \sim 1-\alpha_\ell(s_{j\ell})}[\mathrm{Err}(\zeta)]$.

Now we plugin $T = 2n$, $G = \frac{1}{L} \cdot$ Err, $X_j$s being $\zeta_{j\ell}$s in Theorem C.2. We obtain that for any $t > 0$,

$$\Pr_{\zeta_{j\ell}}\left[\mathrm{Err}(\zeta) \geq \mathrm{Err}_{\mathrm{avg}}(\mu_1, \sigma_1, \mu_2, \sigma_2) + t\right] = \Pr_{\zeta_{j\ell}}\left[\frac{1}{L}\cdot\mathrm{Err}(\zeta) \geq \frac{1}{L}\cdot\mathrm{Err}_{\mathrm{avg}}(\mu_1, \sigma_1, \mu_2, \sigma_2) + \frac{t}{L}\right]$$

$$\leq e^{-\frac{2t^2}{L^2 \mathrm{sg}(\mu_1, \sigma_1, \mu_2, \sigma_2)}},$$

and

$$\Pr_{\zeta_{j\ell}}\left[\mathrm{Err}(\zeta) \leq \mathrm{Err}_{\mathrm{avg}}(\mu_1, \sigma_1, \mu_2, \sigma_2) - t\right] \leq e^{-\frac{2t^2}{L^2 \mathrm{sg}(\mu_1, \sigma_1, \mu_2, \sigma_2)}}. \tag{2}$$

Note that $\mathrm{MaxDisp}_{\mu_1}(\sigma_1, \mu_2, \sigma_2) \geq \mathrm{sg}(\mu_1, \sigma_1, \mu_2, \sigma_2)$. Thus, when $t = L \cdot \sqrt{\mathrm{MaxDisp}_{\mu_1}(\sigma_1, \mu_2, \sigma_2) \cdot \ln\frac{1}{\theta}}$, we have

$$\Pr_{\zeta_{j\ell}}\left[\mathrm{Err}(\zeta) \geq \mathrm{Err}_{\mathrm{avg}}(\mu_1, \sigma_1, \mu_2, \sigma_2) + t\right] \leq \theta \text{ and } \Pr_{\zeta_{j\ell}}\left[\mathrm{Err}(\zeta) \leq \mathrm{Err}_{\mathrm{avg}}(\mu_1, \sigma_1, \mu_2, \sigma_2) - t\right] \leq \theta.$$

This implies that if $\mathrm{Err}_{\mathrm{avg}}(\mu_1, \sigma_1, \mu_2, \sigma_2) \leq \tau - t$,

$$\Pr_{\zeta_{j\ell}}\left[\mathrm{Err}(\zeta) \leq \tau\right] \geq 1 - \Pr_{\zeta_{j\ell}}\left[\mathrm{Err}(\zeta) > \mathrm{Err}_{\mathrm{avg}}(\mu_1, \sigma_1, \mu_2, \sigma_2) + t\right] \geq 1 - \theta. \tag{3}$$

Also, if $\mathrm{Err}_{\mathrm{avg}}(\mu_1, \sigma_1, \mu_2, \sigma_2) \geq \tau + t$,

$$\Pr_{\zeta_{j\ell}}\left[\mathrm{Err}(\zeta) \leq \tau\right] \leq \Pr_{\zeta_{j\ell}}\left[\mathrm{Err}(\zeta) > \mathrm{Err}_{\mathrm{avg}}(\mu_1, \sigma_1, \mu_2, \sigma_2) - t\right] \leq \theta. \tag{4}$$

Recall that $\gamma_1 := \frac{L\sqrt{\mathrm{MaxDisp}_{\mu_1}(\sigma_1, \mu_2, \sigma_2)\cdot\ln(1/\theta)}}{\mathrm{MinDer}_{\mu_1}(\sigma_1, \mu_2, \sigma_2)} = \frac{t}{\mathrm{MinDer}_{\mu_1}(\sigma_1, \mu_2, \sigma_2)}$. Then it suffices to prove the following lemma.

**Lemma C.3.** *Let $t > 0$. Under Assumption 3.1, if $\mu_1 \leq \mu_1^c - \frac{t}{\mathrm{MinDer}_{\mu_1}(\sigma_1, \mu_2, \sigma_2)}$, then $\mathrm{Err}_{avg}(\mu_1, \sigma_1, \mu_2, \sigma_2) \geq \tau + t$; and if $\mu_1 \geq \mu_1^c + \frac{t}{\mathrm{MinDer}_{\mu_1}(\sigma_1, \mu_2, \sigma_2)}$, then $\mathrm{Err}_{avg}(\mu_1, \sigma_1, \mu_2, \sigma_2) \leq \tau - t$.*

*Proof.* We first prove that $\frac{\partial \mathrm{Err}_{\mathrm{avg}}}{\partial \mu_1}(\mu_1, \sigma_1, \mu_2, \sigma_2) \leq 0$. It suffices to prove that for any $\mu, \mu'$ with $\mu \geq \mu'$, $\mathrm{Err}_{\mathrm{avg}}(\mu, \sigma_1, \mu_2, \sigma_2) \leq \mathrm{Err}_{\mathrm{avg}}(\mu', \sigma_1, \mu_2, \sigma_2)$. Let $\alpha_1$ be parameterized by $(\mu, \sigma_1)$, $\alpha_1'$ be parameterized by $(\mu', \sigma_1)$, and $\alpha_2$ be parameterized by $(\mu_2, \sigma_2)$. By Proposition B.1, we know that $\alpha_1'$ has stochastic dominance over $\alpha_1$. Thus, for every $j \in [n]$, there exists a coupling of $(\zeta, \zeta')$ for $\zeta \sim 1-\alpha_1(s_{j\ell})$ and $\zeta' \sim 1-\alpha_1'(s_{j\ell})$ such that $\zeta \leq \zeta'$. Let $\pi_j$ be the joint probability density function of $(\zeta, \zeta')$. We have $\int_{\zeta'} \pi_j(\zeta, \zeta')d\zeta = \alpha_1(s_{j1})(\zeta)$ and $\int_\zeta \pi_j(\zeta, \zeta')d\zeta = \alpha_1'(s_{j1})(\zeta)$. Let $\pi_j$ be the joint probability density function of $(\zeta_{j1}, \zeta_{j1}')$. We have $\int_{\zeta'} \pi_j(\zeta, \zeta')d\zeta = \alpha_1(s_{j1})(\zeta)$ and $\int_\zeta \pi_j(\zeta, \zeta')d\zeta = \alpha_1'(s_{j1})(\zeta)$. Then we have

$$
\begin{aligned}
\mathrm{Err}_{\mathrm{avg}}(\mu, \sigma_1, \mu_2, \sigma_2) =\ & \mathbb{E}_{\zeta_{j\ell}\sim 1-\alpha_\ell(s_{j\ell})}[\mathrm{Err}(\zeta)] \\
=\ & \int \prod_j \pi_j(\zeta_{j\ell}, \zeta_{j\ell}')\mathrm{Err}(\zeta)d\zeta_{j\ell} && \text{(Assumption 3.1 and Defn. of } \pi_j) \\
\leq\ & \int \prod_j \pi_j(\zeta_{j\ell}, \zeta_{j\ell}')\mathrm{Err}(\zeta')d\zeta_{j\ell} && \text{(Montonicity of Err)} \\
\leq\ & \mathbb{E}_{\zeta_{j1}\sim 1-\alpha_1'(s_{j1}), \zeta_{j2}\sim 1-\alpha_2(s_{j2})}[\mathrm{Err}(\zeta)] && \left(\int_\zeta \pi_j(\zeta, \zeta')d\zeta = \alpha_1'(s_{j1})(\zeta)\right) \\
\leq\ & \mathrm{Err}_{\mathrm{avg}}(\mu', \sigma_1, \mu_2, \sigma_2).
\end{aligned}
$$

Thus, if $\mu_1 < \mu_1^c - \frac{t}{\text{MinDer}_{\mu_1}(\sigma_1,\mu_2,\sigma_2)}$, we have

$$
\begin{aligned}
\text{Err}_{\text{avg}}(\mu_1,\sigma_1,\mu_2,\sigma_2) &\geq \text{Err}_{\text{avg}}(\mu_1^c,\sigma_1,\mu_2,\sigma_2) + (\mu_1^c - \mu_1) \cdot \text{MinDer}_{\mu_1}(\sigma_1,\mu_2,\sigma_2) & (\text{Defn. of } \text{MinDer}_{\mu_1}(\sigma_1,\mu_2,\sigma_2)) \\
&\geq \tau + \frac{t}{\text{MinDer}_{\mu_1}(\sigma_1,\mu_2,\sigma_2)} \cdot \text{MinDer}_{\mu_1}(\sigma_1,\mu_2,\sigma_2) & (\text{Err}_{\text{avg}}(\mu_1^c,\sigma_1,\mu_2,\sigma_2) = \tau) \\
&= \tau + t.
\end{aligned}
$$

Similarly, we can prove that if $\mu_1 > \mu_1^c + \frac{t}{\text{MinDer}_{\mu_1}(\sigma_1,\mu_2,\sigma_2)}$, then $\text{Err}_{\text{avg}}(\mu_1,\sigma_1,\mu_2,\sigma_2) < \tau - t$. This completes the proof of Lemma C.3. $\qquad\square$

Combined with Lemma C.3, we have completed the proof.

$\qquad\square$

### C.2. Bounding $\gamma_1$ for alternative choices of error functions and ability profiles

Similar to the illustrative example in Section 3.2, we analyze the window $\gamma_1$ in Theorem 3.2 for alternative choices of the error functions $h, g, f$ and ability profiles $\alpha_1, \alpha_2$. We consider uniform noise such that $\text{MinDer}_{\mu_1}(\sigma_1,\mu_2,\sigma_2) \leq \frac{n\sigma^2}{4}$. Then the key is to bound the Lipschitz constant $L$ and $\text{MinDer}_{\mu_1}(\sigma_1,\mu_2,\sigma_2)$.

**Analysis for** $\max$ **functions.** Let $h, g, f$ be $\max$ such that $\text{Err}(\zeta) = \max_{j\in[n],\ell\in\{1,2\}} \zeta_{j\ell}$. Suppose the ability profile is linear with noise: $\alpha_\ell(s) = 1 - (1 - a_\ell)s + \varepsilon(s)$, where $\varepsilon(s) \sim \text{Unif}[-\sigma,\sigma]$. We can compute that

$$
P = \prod_{j\in[n],\ell\in\{1,2\}} \Pr[\zeta_{j\ell} \leq \tau].
$$

Below, we analyze the window of the phase transition for this case from both the positive and negative sides of the parameter range.

*A positive example.* Assume all $s_{j\ell} = 0.5$, $a_1^c = a_2 = 0.5$, and $\tau = 0.75 + \frac{\sigma}{4} - \frac{\sigma}{4n}$. Then each $\Pr[\zeta_{j\ell} \leq \tau]$ is $1 - \frac{1}{2n}$, implying that

$$
P = \prod_{j\in[n],\ell\in\{1,2\}} \Pr[\zeta_{j\ell} \leq \tau] = (1 - \frac{1}{2n})^{2n} \approx 1/e.
$$

Let $\gamma_1 = 4\sigma/n$. On one hand, if $a_1 = a_1^c - \gamma_1$, we can compute that for each $j \in [n]$, $\Pr[\zeta_{j1} \leq \tau] \leq 1 - \frac{2}{n}$. Then

$$
P \leq (1 - \frac{1}{2n})^n \cdot (1 - \frac{2}{n})^n \leq 1/e^2 < 0.14.
$$

On the other hand, if $a_1 = a_1^c + \gamma_1$, $\Pr[\zeta_{j1} \leq \tau] = 1$ for each $j \in [n]$. Then

$$
P = (1 - \frac{1}{2n})^n \geq 0.6.
$$

Thus, increasing $a_1$ from below $a_1^c - \gamma_1$ to above $a_1^c + \gamma_1$ results in a probability gain of 0.46. The window of this phase transition is only $\gamma_1 = O(\sigma/n)$, which is even sharper than the $O(\sigma/\sqrt{n})$ window observed for the average error function.

*A negative example.* Assume all $s_{j\ell} = 0$ except that $s_{11} = 1$, $a_1^c = a_2 = 0.5$, $\sigma = 0.25$ and $\tau = 0.5$. Then

$$
P = \prod_{j\in[n],\ell\in\{1,2\}} \Pr[\zeta_{j\ell} \leq \tau] = \Pr[\zeta_{11} \leq 0.5] = 0.5.
$$

We can also compute that for $a_1 \in (0.35, 0.65)$,

$$
P = \Pr[\zeta_{11} \leq 0.5] = 0.5 - \frac{a_1 - 0.5}{\min\{a_1, 1 - a_1\}},
$$

which is close to a linear function of $a_1$. Then the window of phase transition is $O(1)$, yielding a smooth, non-abrupt transition.

**Analysis for weighted average functions.** We still select the skill error function $h(\zeta_1, \zeta_2) = \frac{1}{2}(\zeta_1 + \zeta_2)$ as the average function. Given an importance vector $w \in [0,1]^n$ for skills (e.g., derived from O*NET), we select the task error function

$$g(\{h_j\}_{j \in T_i}) = \frac{1}{\sum_{j \in T_j} w_j} \sum_{j \in T_j} w_j h_j,$$

to be the weighted average function, where $\frac{1}{\sum_{j \in T_j} w_j}$ is a normalization factor. Give an importance vector $v \in [0,1]^m$ for tasks (e.g., derived from O*NET), we select the job error function

$$f(g_1, \ldots, g_m) = \frac{1}{\sum_{i \in [m]} v_i} \sum_{i \in [m]} v_i g_i,$$

to be the weighted average function, where $\frac{1}{\sum_{i \in [m]} v_i} \sum_{i \in [m]}$ is a normalization factor. Then we have that their composition function is:

$$\mathsf{Err}(\zeta) = \sum_{j \in [n]} \frac{1}{2} \sum_{i \in [m]: j \in T_i} \frac{v_i}{\sum_{i' \in [m]} v_{i'}} \frac{w_j}{\sum_{j' \in [T_i]} w_{j'}} (\zeta_{j1} + \zeta_{j2}). \tag{5}$$

We have the following observation for the Lipschitzness of $\mathsf{Err}$.

**Proposition C.4 (Lipschitzness of $\mathsf{Err}$ in Equation (5)).** *The lipschitzness constant of $\mathsf{Err}$ in Equation (5) is $L \leq \frac{1}{2} \max_{j \in [n]} \sum_{i \in [m]: j \in T_i} \frac{v_i}{\sum_{i' \in [m]} v_{i'}} \frac{w_j}{\sum_{j' \in [T_i]} w_{j'}}$.*

For instance, when each $w_j = \frac{1}{n}$ and $v_i = \frac{1}{m}$, we have $L \leq \frac{1}{2} \max_{j \in [n]} \sum_{i \in [m]: j \in T_i} \frac{1}{m|T_i|}$. Specifically, when each $T_i$ contains $k$ skills and each skill appears in $\frac{km}{n}$ tasks, we have $L \leq \frac{1}{2n}$. Suppose the ability profile is linear with noise: $\alpha_\ell(s) = 1 - (1 - a_\ell)s + \varepsilon(s)$, where $\varepsilon(s) \sim \mathrm{Unif}[-\sigma, \sigma]$. Similarly, we can compute that

$$\mathrm{MinDer}_{a_1}(\sigma, a_2, \sigma) = \frac{1}{2} \sum_{j \in [n]} \sum_{i \in [m]: j \in T_i} \frac{v_i}{\sum_{i' \in [m]} v_{i'}} \frac{w_j}{\sum_{j' \in [T_i]} w_{j'}} s_{j1}.$$

Then we have the following bound for $\gamma_1$:

$$\gamma_1 = \frac{L \sqrt{0.5 n \sigma^2 \cdot \ln(1/\theta)}}{\mathrm{MinDer}_{\mu_1}(\sigma_1, \mu_2, \sigma_2)} \leq \frac{\max_{j \in [n]} \sum_{i \in [m]: j \in T_i} \frac{v_i}{\sum_{i' \in [m]} v_{i'}} \frac{w_j}{\sum_{j' \in [T_i]} w_{j'}} \cdot \sqrt{0.5 n \sigma^2 \cdot \ln(1/\theta)}}{\sum_{j \in [n]} \sum_{i \in [m]: j \in T_i} \frac{v_i}{\sum_{i' \in [m]} v_{i'}} \frac{w_j}{\sum_{j' \in [T_i]} w_{j'}} s_{j1}}.$$

**Analysis for constant profiles.** We select $\alpha_\ell = c_\ell + \min\{c_\ell, 1 - c_\ell\} \mathrm{Unif}[-\sigma, \sigma]$ as constant profiles with noise level $\sigma$ as detailed in Section B.1. We still let $\mathsf{Err}(\zeta) = \frac{1}{2n} \sum_{j=1}^n (\zeta_{j1} + \zeta_{j2})$, where $L \leq \frac{1}{2n}$. Note that $\mathrm{MinDer}_{c_1}(\sigma, c_2, \sigma) = \frac{1}{2}$. Then we have the following bound for $\gamma_1$:

$$\gamma_1 = \frac{L \sqrt{0.5 n \sigma^2 \cdot \ln(1/\theta)}}{\mathrm{MinDer}_{c_1}(\sigma, c_2, \sigma)} \leq \sigma \cdot \sqrt{\frac{\ln(1/\theta)}{2n}}.$$

**Analysis for polynomial profiles.** We select $\alpha_\ell \equiv 1 - s^{\beta_\ell} + \min\{s^{\beta_\ell}, 1 - s^{\beta_\ell}\} \mathrm{Unif}[-\sigma, \sigma]$ as polynomial profiles with noise level $\sigma$. We still let $\mathsf{Err}(\zeta) = \frac{1}{2n} \sum_{j=1}^n (\zeta_{j1} + \zeta_{j2})$, where $L \leq \frac{1}{2n}$. Then we have

$$\left| \frac{\partial \mathsf{Err}_{\mathrm{avg}}}{\partial \beta_1} (\beta_1, \sigma, \beta_2, \sigma) \right| = \frac{\beta_1}{2n} \sum_{j=1}^n s_{j1}^{\beta_1 - 1}.$$

Note that this partial derivative is 0 when $\beta_1 = 0$, which results in $\mathrm{MinDer}_{\beta_1}(\sigma, \beta_2, \sigma) = 0$ and $\gamma_1 = \infty$.

However, by the proof of Theorem C.1, it suffices to bound the partial derivative for $\beta_1 \in [\beta_1^c - \gamma_1, \beta_1^c + \gamma_1]$ instead of the entire domain $\mathbb{R}_{\geq 0}$. Suppose we know that $[\beta_1^c - \gamma_1, \beta_1^c + \gamma_1] \subseteq [0.5, 2]$; this implies that

$$\frac{\beta_1}{2n} \sum_{j=1}^n s_{j1}^{\beta_1 - 1} \geq \frac{1}{4n} \sum_{j=1}^n s_{j1}.$$

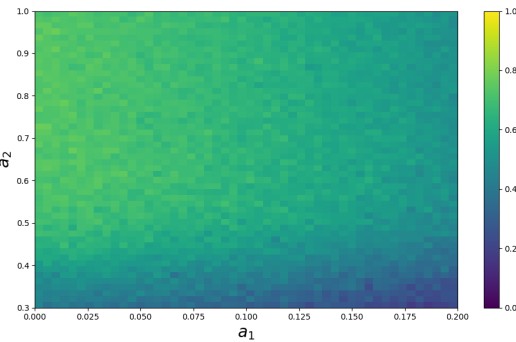

*Figure 6.* Heatmaps of productivity compression value $\text{PC} = (P_2 - P_1) - (P_2' - P_1')$ by merging a low-skilled human worker with action-level ability parameter $a_1$ and a high-skilled human worker with action-level ability parameter $a_2$ with a GenAI tool for different ranges of $(a_1, a_2)$ for the Computer Programmers example with default settings of $\tau = 0.45$.

Thus, we have the following bound for $\gamma_1$:

$$\gamma_1 = \frac{L\sqrt{0.5n\sigma^2 \cdot \ln(1/\theta)}}{\frac{1}{4n}\sum_{j=1}^{n} s_{j1}} \leq \sigma \cdot \sqrt{\frac{2n \cdot \ln(1/\theta)}{\sum_{j=1}^{n} s_{j1}}}.$$

### C.3. Proof of Theorem 3.3: Success gain from merging complementary workers

Similar to Section C.1, we extend Theorem 3.3 to handle a general noise model $\varepsilon(s)$. The only difference is still the introduction of MaxDisp.

**Theorem C.5** (**Extension of Theorem 3.3 to general noise models**). *Fix the job instance. Let $\theta \in (0, 0.5)$ be a confidence level, and define:* $\gamma_1^{(1)} := \frac{L \cdot \sqrt{\text{MaxDisp}_{\mu_1}(\sigma_1^{(1)}, \mu_2^{(2)}, \sigma_2^{(2)}) \cdot \ln(1/\theta)}}{\text{MinDer}_{\mu_1}(\sigma_1^{(1)}, \mu_2^{(2)}, \sigma_2^{(2)})}$, *and* $\gamma_1^{(2)} := \frac{L \cdot \sqrt{\text{MaxDisp}_{\mu_1}(\sigma_1^{(2)}, \mu_2^{(2)}, \sigma_2^{(2)}) \cdot \ln(1/\theta)}}{\text{MinDer}_{\mu_1}(\sigma_1^{(2)}, \mu_2^{(2)}, \sigma_2^{(2)})}$. *If*

$$\text{Err}_{avg}(\mu_1^{(1)} - \gamma_1^{(1)}, \sigma_1^{(1)}, \mu_2^{(2)}, \sigma_2^{(2)}) \leq \tau \leq \text{Err}_{avg}(\mu_1^{(2)} + \gamma_1^{(2)}, \sigma_1^{(2)}, \mu_2^{(2)}, \sigma_2^{(2)}),$$

*then under Assumption 3.1, we have:* $P_{12} - P_2 \geq 1 - 2\theta$.

*Proof.* If $\text{Err}_{\text{avg}}(\mu_1^{(2)} + \gamma_1^{(2)}, \sigma_1^{(2)}, \mu_2^{(2)}, \sigma_2^{(2)}) \geq \tau$, by the proof of Theorem 3.2, we know that

$$\text{Err}_{\text{avg}}(\mu_1^{(2)}, \gamma_1^{(2)}, \mu_2^{(2)}, \sigma_2^{(2)}) \geq \tau + \gamma_1^{(2)}\text{MinDer}_{\mu_1}(\sigma_1^{(2)}, \mu_2^{(2)}, \sigma_2^{(2)}) = \tau + L \cdot \sqrt{\text{MaxDisp}_{\mu_1}(\sigma_1^{(2)}, \mu_2^{(2)}, \sigma_2^{(2)}) \cdot \ln(1/\theta)}.$$

By Inequality (4), we conclude that $P_2 \leq \theta$. Similarly, by $\text{Err}_{\text{avg}}(\mu_1^{(1)} - \gamma_1^{(1)}, \sigma_1^{(1)}, \mu_2^{(2)}, \sigma_2^{(2)}) \leq \tau$ and Inequality (3), we can obtain $P_{12} \geq 1 - \theta$. Thus, $\Delta_2 \geq 1 - 2\theta$, which completes the proof. $\square$

### C.4. Proof of Corollary 3.4 and extension to distinct ability profiles

Similar to Section C.1, we extend Theorem 3.3 to handle a general noise model $\varepsilon(s)$. The only difference is still the introduction of MaxDisp.

**Corollary C.6** (**Extension of Corollary 3.4 to general noise models**). *Fix the job instance. Suppose both human workers have the same decision-level abilities:*

$$\mu_1^{(1)} = \mu_1^{(2)} = \mu_1^\star > \mu_1^{(\text{AI})}, \quad \sigma_1^{(1)} = \sigma_1^{(2)} = \sigma_1^{(\text{AI})} = \sigma_1^\star.$$

*Let $\theta \in (0, 0.5)$ be a confidence level, and for each $\ell \in \{1, 2, \text{AI}\}$, define* $\gamma_2^{(\ell)} := \frac{L \cdot \sqrt{\text{MaxDisp}_{\mu_2}(\sigma_2^{(\ell)}, \mu_1^\star, \sigma_2^\star) \cdot \ln(1/\theta))}}{\text{MinDer}_{\mu_2}(\sigma_2^{(\ell)}, \mu_1^\star, \sigma_1^\star)}$. *If*

$$\max\left\{\text{Err}_{avg}(\mu_1^\star, \sigma_1^\star, \mu_2^{(\text{AI})} - \gamma_2^{(\text{AI})}, \sigma_2^{(\text{AI})}), \text{Err}_{avg}(\mu_1^\star, \sigma_1^\star, \mu_2^{(2)} - \gamma_2^{(2)}, \sigma_2^{(2)})\right\} \leq \tau \leq \text{Err}_{avg}(\mu_1^\star, \sigma_1^\star, \mu_2^{(1)} + \gamma_2^{(1)}, \sigma_2^{(1)}),$$

*then under Assumption 3.1, we have:* $\text{PC} \geq 1 - 2\theta$.

*Proof.* By the assumption on the decision-level abilities, the merging of $W_\ell$ and $W_{\mathrm{AI}}$ must utilize a decision-level ability profile parameterized by $(\mu_1^\star, \sigma_1^\star)$. Since $\mathsf{Err}_{\mathrm{avg}}(\mu_1^\star, \sigma_1^\star, \mu_2^{(2)} - \gamma_2^{(2)}, \sigma_2^{(2)}) \leq \tau \leq \mathsf{Err}_{\mathrm{avg}}(\mu_1^\star, \sigma_1^\star, \mu_2^{(1)} + \gamma_2^{(1)}, \sigma_2^{(1)})$, it follows from Theorem 3.3 that $P_2 - P_1 \geq 1 - 2\theta$. Also, since $\mathsf{Err}_{\mathrm{avg}}(\mu_1^\star, \sigma_1^\star \sigma_1^\star, \mu_2^{(AI)} - \gamma_2^{(AI)}, \sigma_2^{(AI)}) \leq \tau \leq \mathsf{Err}_{\mathrm{avg}}(\mu_1^\star, \mu_2^{(1)} + \gamma_2^{(1)}, \sigma_2^{(1)})$, it follows from Theorem 3.3 that $P_1' - P_1 \geq 1 - 2\theta$. Also note that $P_1' \leq P_2'$. Hence,

$$\mathrm{PC} = |P_2 - P_1| - |P_2' - P_1'| = P_2 - P_1 + P_1' - P_2'.$$

If the merging of $W_2$ and $W_{\mathrm{AI}}$ utilizes $W_2$'s action-level abilities, we have $P_2 = P_2'$ and hence,

$$\mathrm{PC} = P_2 - P_1 + P_1' - P_2' = P_1' - P_1 \geq 1 - 2\theta.$$

Otherwise, if the merging of $W_2$ and $W_{\mathrm{AI}}$ utilizes $W_{\mathrm{AI}}$'s action-level abilities, we have $P_2' = P_1'$ and hence,

$$\mathrm{PC} = P_2 - P_1 + P_1' - P_2' = P_2 - P_1 \geq 1 - 2\theta.$$

Overall, we have completed the proof. $\qquad\square$

**Evaluating productivity compression with distinct ability profiles.** Similar to Section 4, we investigate whether the productivity compression effect induced by AI assistance persists when the ability profiles of human workers and GenAI originate from different functional families.

*Choice of parameters.* We set the decision-level ability profiles of $W_1$ and $W_2$ to be linear with $\alpha_1^{(1)}(s) = \alpha_1^{(2)}(s) = \mathrm{TrunN}(1 - 0.78s, 0.0065; 0, 1)$, and define their action-level ability profiles as $\alpha_\ell^{(2)}(s) = \mathrm{TrunN}(1 - (1 - a_\ell)s, 0.0065; 0, 1)$. We assume $a_2 > a_1$, representing two human workers with distinct skill levels. The parameter ranges are set as $a_1 \in [0, 0.2]$ and $a_2 \in [0.3, 1]$, motivated by the observation that $a = 0.22$ corresponds to a job success probability of $0.55$, characterizing a medium-skilled worker. For the GenAI tool $W_{\mathrm{AI}}$, we define the decision-level ability as $\alpha_1^{(AI)}(s) = \mathrm{TrunN}(1 - 0.92s, 0.0145; 0, 1)$ and the action-level ability as $\alpha_2^{(AI)}(s) = \mathrm{TrunN}(0.8, 0.0145; 0, 1)$, such that the decision-level ability is consistently weaker than that of $W_\ell$. We adopt the same merging scheme between human workers $W_\ell$ and the GenAI tool $W_{\mathrm{AI}}$. Recall that for $\ell \in \{1, 2\}$, $P_\ell$ denotes the job success probability of $W_\ell$ before merging with $W_{\mathrm{AI}}$, and $P_\ell'$ denotes the corresponding probability after merging.

*Analysis.* Figure 6 presents a heatmap of $\mathrm{PC} := (P_2 - P_1) - (P_2' - P_1')$ as the ability parameters $a_1$ and $a_2$ vary. We observe that PC increases with the ability gap $a_2 - a_1$, indicating that the benefit of merging is more pronounced for lower-skilled workers. For instance, when $a_1 = 0.1$ and $a_2 = 0.8$, the productivity compression reaches $\mathrm{PC} = 0.8$. These findings confirm that the productivity compression effect from human-AI collaboration persists even when workers specialize in different action-level subskills, thereby affirming our hypothesis.

## C.5. Extending Theorem 3.2 to noise-dependent settings

We consider the noise-dependent setting introduced in Section 4. In this setting, a dependency parameter $p \in [0, 1]$ controls whether subskill errors are drawn from a shared latent factor $\beta$ (with probability $p$) or independently (with probability $1 - p$). The following theorem extends Theorem 3.2, which corresponds to the independent case $p = 0$, to general $p \in [0, 1]$. Notably, the sensitivity window $\gamma_1$ vanishes as $p \to 1$, indicating that stronger dependencies smooth out abrupt transitions.

**Theorem C.7** (**Phase transition in noise-dependent settings**). *Fix the job instance, action-level ability $\mu_2$, and noise levels $\sigma_1, \sigma_2$. Let $\mu_1^c$ be the unique value such that the expected job error equals the success threshold:*

$$\mathsf{Err}_{avg}(\mu_1^c, \sigma_1, \mu_2, \sigma_2) = \tau.$$

*Let $\theta \in (0, 0.5)$ be a confidence level, $p \in [0, 1]$ be a dependency parameter, and define the transition width: $\gamma_1 := \frac{L\sqrt{\mathrm{MaxDisp}_{\mu_1}(\sigma_1, \mu_2, \sigma_2)} \cdot \max\{\sqrt{\ln \frac{2(1-p)}{\theta}}, \sqrt{n \ln \frac{2p}{\theta}}\}}{\mathrm{MinDer}_{\mu_1}(\sigma_1, \mu_2, \sigma_2)}$, where $L$ is the Lipschitz constant of the job error function. Then the job success probability satisfies:*

$$P \leq \theta \ \text{ if } \ \mu_1 \leq \mu_1^c - \gamma_1 \ \text{and} \ P \geq 1 - \theta \ \text{ if } \ \mu_1 \geq \mu_1^c + \gamma_1.$$

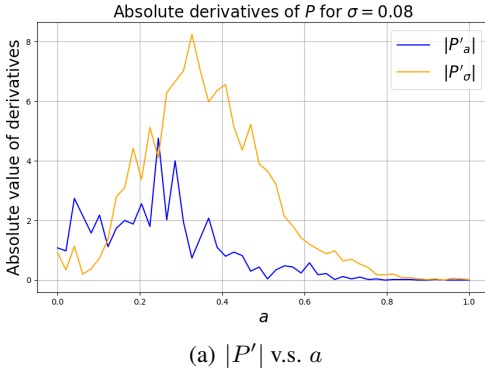
(a) $|P'|$ v.s. $a$

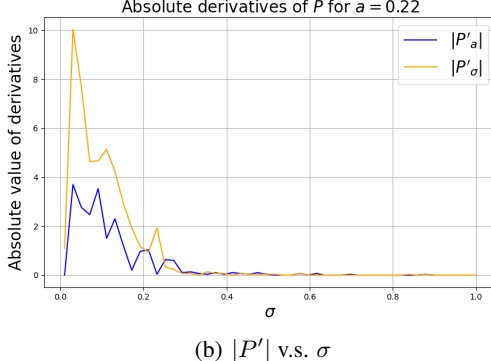
(b) $|P'|$ v.s. $\sigma$

*Figure 7.* Plots illustrating the relationship between the absolute derivatives $|P'_a|$ and $|P'_\sigma|$ and $a$, $\sigma$ for the Computer Programmers example with default settings of $(a, \sigma, \tau) = (0.22, 0.08, 0.45)$.

*Proof.* The main difference is that the probability bound by Inequality (2) in the proof of Theorem 3.2 changes to be:

$$\Pr_{\zeta_{j\ell}}\left[\mathsf{Err}(\zeta) \leq \mathsf{Err}_{\mathrm{avg}}(\mu_1, \sigma_1, \mu_2, \sigma_2) - t\right] \leq (1-p) \cdot e^{-\frac{2t^2}{L^2 \mathrm{sg}(\mu_1, \sigma_1, \mu_2, \sigma_2)}} + p \cdot e^{-\frac{2t^2}{L^2 n \cdot \mathrm{sg}(\mu_1, \sigma_1, \mu_2, \sigma_2)}}.$$

The choice of $\gamma_1$ ensures the right-hand side to be at most $\theta$, which completes the proof. $\square$

# D. Additional implications of theoretical results

We empirically analyze the impact of workers' ability profiles on job success probability. In Section D.1, we illustrate how our framework can be used to determine strategies to upskill workers. In Section D.2, we analyze the impact of evaluation bias on workers' abilities.

## D.1. Evaluating intervention effectiveness: Boosting ability v.s. reducing noise

As discussed in Section 3 (see also Figure 5(b)), both increasing the ability parameter and reducing the noise level are efficient interventions for increasing the job success probability. We consider the "Computer Programmers" example with independent abilities across subskills ($p = 1$) in Section 4. Then, the job error function $\mathsf{Err}$ is defined as in Equation (13) and the subskill numbers are as in Equation (11). We set the subskill ability profiles to be $\alpha_1^{(1)}(s) = \mathrm{TrunN}(1 - (1-a)s, \sigma^2/2; 0, 1)$ and $\alpha_2^{(1)}(s) = \mathrm{TrunN}(1 - 0.78s, \sigma^2/2; 0, 1)$, representing a human worker. We investigate which parameter-$a, \sigma$-has the greatest impact on $P$. This analysis is crucial for guiding strategies to upskill workers for specific jobs. To this end, we first compute the derivatives of $P$ with respect to $a$ and $\sigma$. We denote $|P'_a|$ and $|P'_\sigma|$ as the absolute values of the derivative of $P$ with respect to $a$ and $\sigma$, respectively. We plot them in Figure 7 for the default parameters $(a, \sigma, \tau) = (0.22, 0.08, 0.45)$.

Figure 7(a) reveals that for $\sigma = 0.08$, when $a \in [0, 0.15]$, $|P'_a|$ is larger; while when $a \in [0.15, 1]$, $|P'_\sigma|$ is larger. In Figure 7(b), for $a = 0.22$, $|P'_\sigma|$ is always larger for any $\sigma \in [0, 1]$. Thus, in this specific example, depending on the ranges of $(a, \sigma)$, either $|P'_a|$ or $|P'_\sigma|$ may be larger, demonstrating that no single parameter universally outweighs the others in importance.

## D.2. Analyzing the impact of inaccurate ability evaluation

We demonstrate how Theorem 3.2 highlights the importance of accurately evaluating workers' abilities for companies. Specifically, we consider the scenario where a worker's ability evaluations are biased and discuss the consequences of this bias. Mathematically, let the worker's true decision-level ability parameter be $\mu_1$, while the observed parameter is $\widehat{\mu_1} = \beta \mu_1$ for some $\beta \in (0, 1)$, reflecting bias in the evaluation process. This bias model is informed by and builds upon a substantial body of research on selection processes in biased environments (Kleinberg & Raghavan, 2018; Celis et al., 2020). Below, we quantify the impact of such bias $\beta$ on workers.

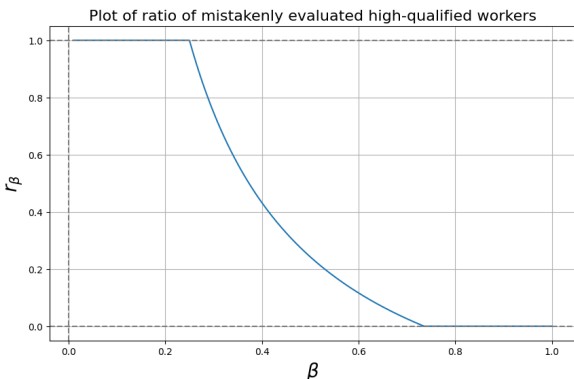

*Figure 8.* Plots illustrating the relationship between the ratio $r_\beta$ and the bias parameter $\beta$ for the Computer Programmers example with default settings of $\tau = 0.45$.

**Theoretical analysis.** Suppose parameters $\mu_\ell^\star, \sigma_\ell^\star$ satisfy that $\mathsf{Err}_{\mathrm{avg}}(\mu_1^\star, \sigma_1^\star, \mu_2^\star, \sigma_2^\star) = \tau$. Let $\theta \in (0, 0.5)$. Let $\gamma_1 :=$ $\frac{L \cdot \sqrt{\mathrm{MaxDisp}_{\mu_1}(\sigma_1, \mu_2, \sigma_2) \cdot \ln \frac{1}{\theta}}}{\mathrm{MinDer}_{\mu_1}(\sigma_1, \mu_2, \sigma_2)}$. According to Theorem 3.2, if $\mu_1 \geq \mu_1^\star + \gamma_1$, the job success probability $P(\alpha_1, \alpha_2, h, g, f, \tau) \geq 1 - \theta$, indicating that the worker fits the job. However, the evaluated success probability $\widehat{P}$ is based on a weaker ability profile $\widehat{\alpha}$, parameterized by $(\widehat{\mu_1}, \sigma_1)$. From Theorem 3.2, if $\widehat{\mu_1} \leq \mu_1^\star - \gamma_1$, the evaluated success probability $\widehat{P} \leq \theta$, implying that the evaluation process concludes the worker does not fit the job. Since $\widehat{\mu_1} = \beta \mu_1$, we conclude that $\widehat{P} \leq \theta$ if $\mu_1 \leq \frac{1}{\beta}(\mu_1^\star - \gamma_1)$. Note that $\frac{1}{\beta}(\mu_1^\star - \gamma_1) > \mu_1^\star + \gamma_1$ when $\beta < \frac{\mu_1^\star - \gamma_1}{\mu_1^\star + \gamma_1}$. Recall that $\gamma_1 = O(\sigma \sqrt{\frac{\ln \frac{1}{\theta}}{n}})$ in the linear ability example shown in Section 3.3, which is $o(1)$ when $\sigma \ll \frac{1}{\sqrt{\ln \frac{1}{\theta}}}$ or $n \gg \ln \frac{1}{\theta}$. Thus, the condition $\beta < \frac{\mu_1^\star - \gamma_1}{\mu_1^\star + \gamma_1}$ is $\beta < 1 - o(1)$. Consequently, for workers with ability parameter $\mu_1 \in [\mu_1^\star + \gamma_1, \frac{1}{\beta}(\mu_1^\star - \gamma_1)]$, the evaluated success probability $\widehat{P} \leq \theta$, while the true success probability $P \geq 1 - \theta$. Thus, even a slight bias in ability evaluations can lead to dramatic errors in predicting the worker's job success probability, potentially causing companies to lose qualified workers.

**Simulation for one worker.** We study the ratio of high-qualified workers with $P \geq 0.8$ who are mistakenly evaluated as insufficiently qualified with $\widehat{P} \leq 0.6$ due to evaluation bias. Again, we take the example of "Computer Programmers" as an illustration. We set the decision-level ability profile to be $\mathrm{TrunN}(1 - (1-a)s, 0.0065; 0, 1)$ and the action-level ability profile to be $\mathrm{TrunN}(1 - 0.78s, 0.0065; 0, 1)$, representing human workers. We assume the ability parameter $a$ follows from the density $\mathrm{Unif}[0, 1]$ for mathematical simplicity (can be changed to e.g., a truncated normal distribution). By Figure 3(a), we know that $P \geq 0.8$ if $a \geq 0.34$. Thus, 66% of workers are highly qualified for this job. In contrast, $\widehat{P} \leq 0.6$ if $\widehat{a} = \beta a \leq 0.25$, i.e., $a \leq \frac{0.25}{\beta}$. Thus, high-qualified workers with ability parameter $a \in [0.34, \frac{0.25}{\beta}]$ are mistakenly evaluated as insufficiently qualified. Then, the ratio of these workers among high-qualified workers is $r_\beta = \min \left\{ 1, \max \left\{ 0, \frac{0.25}{\beta} - 0.34 \right\} / 0.66 \right\}$; see Figure 8 for a visualization. We note that $r_\beta > 0$ when $\beta \leq 0.64$, and it increases super-linearly to 1 as $\beta$ decreases from 0.64 to 0.25.

**Simulation for merging two workers.** Next, we study how inaccurate ability estimation affects the gain in success probability from the merging process. We extend our merging analysis by introducing a trust parameter $\lambda$ to the merging experiment in Section 4, which models imperfect merging by letting the estimated ability $\widehat{c} = \lambda c$ deviate from the true action-level ability $c$ of worker $W_2$. We then assign action-level subskills to $W_2$ when its scaled ability, $\lambda c$, exceeds $W_1$'s ability (i.e., $1 - 0.78 s_{j2} \leq \lambda c$, even though $W_2$ completes skills at level $c$).

Figure 9 plots the probability gain $\Delta = P_{\mathrm{merge}} - \max\{P_1, P_2\}$ across different values of $c$ and $\lambda$. We find that even modest errors in $\lambda$ can sharply reduce $\Delta$. For example, when $\lambda = 1.14$ and $c = 0.2$, the probability gain becomes $\Delta = -0.2$, indicating that merging reduces job success. This illustrates the critical importance of accurate ability estimation, and complements the findings in Section D.2 on belief-driven merging.

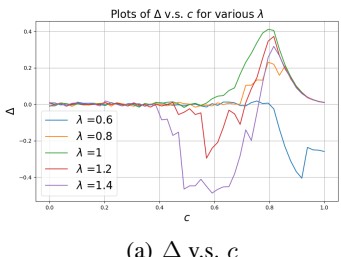
(a) $\Delta$ v.s. $c$

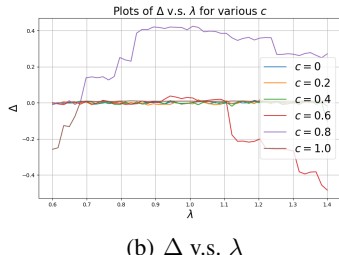
(b) $\Delta$ v.s. $\lambda$

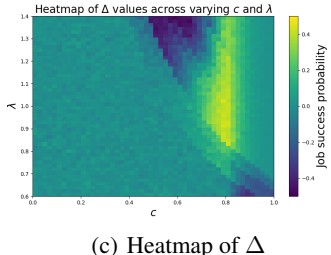
(c) Heatmap of $\Delta$

*Figure 9.* Plots illustrating the relationship between the probability gain $\Delta = P_{merge} - \max\{P_1, P_2\}$ and the $W_2$'s action-level ability parameter $c$ and trust parameter $\lambda$ to $W_2$'s action-level ability for the Computer Programmers example with default settings of $\tau = 0.45$. Here, $\lambda > 1$ indicates an overestimate of $W_2$'s action-level ability, while $\lambda < 1$ indicates an underestimate. We assign action-level subskills to $W_2$ if its evaluated ability $\lambda c$ dominates that of $W_1$, i.e., $1 - 0.78 s_{j2} \leq \lambda c$. Notably, the probability gain $\Delta$ can be negative due to the imperfect merging introduced by the trust parameter.

## E. Omitted details from Section 4

We provide additional details for the examples stated in Section 4.

**O\*NET.** O\*NET, developed by the U.S. Department of Labor, is a comprehensive database providing standardized descriptions of occupations, including required skills, knowledge, abilities, and work activities. It helps job seekers, employers, educators, and policymakers understand workforce needs and trends. O\*NET aids career exploration, job description development, curriculum design, and labor market analysis. Employers use it to identify workforce needs, while educators align training programs with job market demands. O\*NET provides skill proficiency levels but lacks granularity in distinguishing decision-making from action-based abilities. GenAI excels in technical execution but struggles with strategic problem-solving. Enhancing O\*NET to capture these distinctions would improve AI-human job interaction analysis and workforce planning.

### E.1. Details of deriving job data from O\*NET

The job of "Computer Programmer" consists of $n = 18$ skills and $m = 17$ tasks, together with their descriptions (link: https://www.onetonline.org/link/summary/15-1251.00). O\*NET also offers the importance of each task, which may influence the choice of job error function $f$. The task importance vector is

$$v = (.86, .85, .84, .79, .76, .74, .65, .64, .63, .57, .57, .57, .56, .63, .56, .49, .46), \tag{6}$$

where $v_i$ indicates the importance level of task $i$. Additionally, O\*NET offers the importance and proficiency level of each skill. The skill importance vector is

$$w = (.5, .53, .53, .53, .5, .6, .56, .56, .56, .53, .63, .6, .53, .53, .69, .69, .69, .94) \in [0,1]^n, \tag{7}$$

where $w_j$ indicates the importance level of skill $j$, which may influence the choice of task error function $g$. The skill proficiency vector is

$$s = (.41, .43, .45, .45, .45, .46, .46, .46, .46, .48, .5, .5, .52, .54, .55, .55, .57, .7) \in [0,1]^n, \tag{8}$$

where $s_j$ represents the criticality of skill $j$ for this job. We summarize how to derive this data in Figure 10. The derived data for tasks and skills are summarized in Tables 1 and 2, respectively.

### E.2. Details of deriving workers' abilities from Big-bench Lite

We show how to formulate ability profiles for human workers and GenAI tools via skill evaluations in Big-bench Lite (bench authors, 2023). BIG-bench Lite (BBL) is a curated subset of the Beyond the Imitation Game Benchmark (BIG-bench), designed to evaluate large language models efficiently. While BIG-bench contains over 200 diverse tasks, BBL selects 24 representative tasks covering domains such as code understanding, multilingual reasoning, logical deduction, and social bias

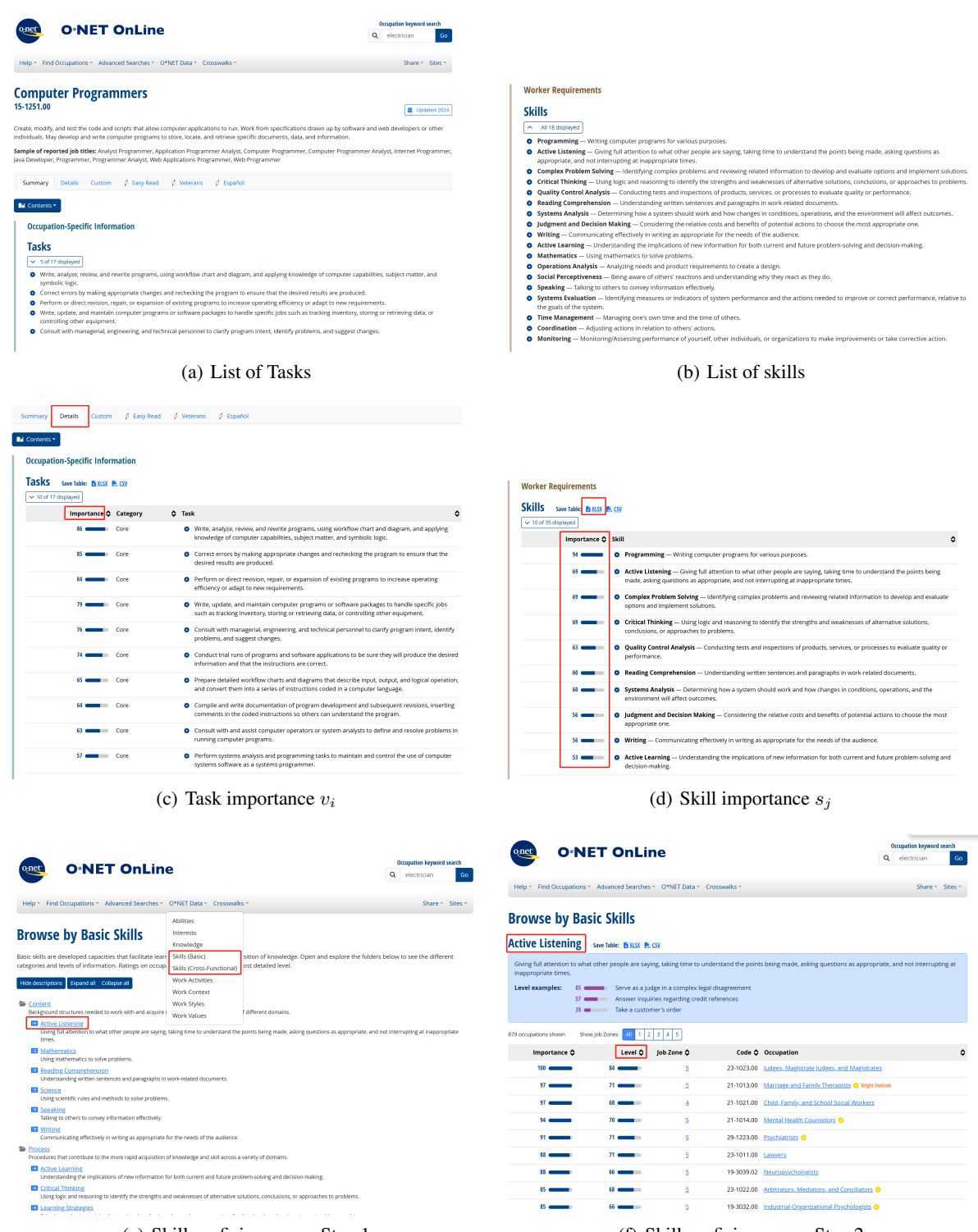

(a) List of Tasks

(b) List of skills

(c) Task importance $v_i$

(d) Skill importance $s_j$

(e) Skill proficiency $w_j$: Step 1

(f) Skill proficiency $w_j$: Step 2

*Figure 10.* Deriving job data for computer programmers from O*NET. Subfigures (a) and (b) are on the "Summary" page of the job (link: https://www.onetonline.org/link/summary/15-1251.00). Subfigures (c) and (d) are on the "Details" page. Subfigures (e) and (f) show how to obtain skill proficiencies $s_j$s from O*NET.

*Table 1.* Data for tasks associated with the job of "Computer Programmers."

| Task id | Task name | Importance ($v\%$) |
|---|---|---|
| 1 | Write, analyze, review, and rewrite programs, using workflow chart and diagram, and applying knowledge of computer capabilities, subject matter, and symbolic logic | 86 |
| 2 | Correct errors by making appropriate changes and rechecking the program to ensure that the desired results are produced | 85 |
| 3 | Perform or direct revision, repair, or expansion of existing programs to increase operating efficiency or adapt to new requirements | 84 |
| 4 | Write, update, and maintain computer programs or software packages to handle specific jobs such as tracking inventory, storing or retrieving data, or controlling other equipment | 79 |
| 5 | Consult with managerial, engineering, and technical personnel to clarify program intent, identify problems, and suggest changes | 76 |
| 6 | Conduct trial runs of programs and software applications to be sure they will produce the desired information and that the instructions are correct | 74 |
| 7 | Prepare detailed workflow charts and diagrams that describe input, output, and logical operation, and convert them into a series of instructions coded in a computer language | 65 |
| 8 | Compile and write documentation of program development and subsequent revisions, inserting comments in the coded instructions so others can understand the program | 64 |
| 9 | Consult with and assist computer operators or system analysts to define and resolve problems in running computer programs | 63 |
| 10 | Perform systems analysis and programming tasks to maintain and control the use of computer systems software as a systems programmer | 57 |
| 11 | Write or contribute to instructions or manuals to guide end users | 57 |
| 12 | Investigate whether networks, workstations, the central processing unit of the system, or peripheral equipment are responding to a program's instructions | 57 |
| 13 | Assign, coordinate, and review work and activities of programming personnel | 56 |
| 14 | Train subordinates in programming and program coding | 63 |
| 15 | Develop Web sites | 56 |
| 16 | Train users on the use and function of computer programs | 49 |
| 17 | Collaborate with computer manufacturers and other users to develop new programming methods | 46 |

*Table 2.* Data for skills associated with the job of "Computer Programmer"; sorted in an increasing order of proficiency.

| Skill id | Skill name | Importance ($w\%$) | Proficiency ($s\%$) | Decomposition ($\lambda$) | Decision ($s_{j1}$) | Action ($s_{j2}$) |
|---|---|---|---|---|---|---|
| 1 | Coordination | 50 | 41 | 0 | 0 | 0.41 |
| 2 | Social Perceptiveness | 53 | 43 | 0 | 0 | 0.43 |
| 3 | Mathematics | 53 | 45 | 1 | 0.45 | 0 |
| 4 | Time Management | 53 | 45 | 1 | 0.45 | 0 |
| 5 | Monitoring | 50 | 45 | 1 | 0.45 | 0 |
| 6 | Systems Analysis | 60 | 45 | 0.6 | 0.27 | 0.18 |
| 7 | Judgment and Decision Making | 56 | 46 | 0.7 | 0.322 | 0.138 |
| 8 | Writing | 56 | 46 | 0.4 | 0.184 | 0.276 |
| 9 | Active Learning | 56 | 46 | 0.4 | 0.184 | 0.276 |
| 10 | Speaking | 53 | 48 | 0 | 0 | 0.48 |
| 11 | Quality Control Analysis | 63 | 50 | 0.3 | 0.15 | 0.35 |
| 12 | Reading Comprehension | 60 | 50 | 1 | 0.5 | 0 |
| 13 | Systems Evaluation | 53 | 52 | 1 | 0.52 | 0 |
| 14 | Operations Analysis | 53 | 54 | 0.6 | 0.324 | 0.216 |
| 15 | Complex Problem Solving | 69 | 55 | 0.7 | 0.385 | 0.165 |
| 16 | Critical Thinking | 69 | 55 | 0.6 | 0.33 | 0.22 |
| 17 | Active Listening | 69 | 57 | 0 | 0 | 0.57 |
| 18 | Programming | 94 | 70 | 0.4 | 0.28 | 0.42 |

assessment. bench authors (2023) assessed the accuracies of the best human rater, the average human rater, and the best LLM for these skills in Big-bench Lite. By Figure 1(c) of (bench authors, 2023), we know that the best LLM refers to PaLM (Chowdhery et al., 2023).

Our goal is to formulate the ability profile of a human worker using the data for the average human rater, and formulate the ability profile for a GenAI tool using the data for the best LLM. Their accuracies for 24 skills are summarized in Table 3. Note that the accuracy corresponds to the average ability of workers. Thus, to formulate ability profiles, we need to know the proficiencies of these skills and the variance in the workers' abilities. Below, we illustrate how to derive this data.

**Deriving skill proficiencies.** We use GPT-4o to derive proficiencies for the 24 skills and obtain a skill proficiency vector in $[0, 1]^{24}$:

$$s = (0, .87, .65, 1, .33, .98, .60, .80, .91, .27, 0, .20, .20, .75, .71, .25, .00, .73, .07, .91, .64, .00, .64, .50).$$

The prompt is: "# Table 3. Given the list of 24 skills in Big-bench Lite, please construct a 24-vector $s$ where $s_j$ represents the proficiency level of skill $j$ with 0 for the easiest and 1 for the hardest."

**Deriving the ability profiles.** Given accuracies in Table 3, we have the accuracies of the average human rater and the best LLM. Combining the accuracies and the skill proficiency vector $s$, we observe that linear functions can fit skill ability profiles of the average human rater and LLM; see Figure 11. We fit the ability of the average human rater by $1 - 0.78s$, whose estimation variance is 0.013. Also, we fit the ability of the LLM by $1 - 0.92s$, whose estimation variance is 0.029. Additionally, by Figure App.9 of (bench authors, 2023), we observe that the noise distribution of abilities is close to a truncated normal distribution. Overall, we formulate the ability profiles of a human worker ($W_1$) and a GenAI tool ($W_2$) to be

$$\alpha^{(1)}(s) = \mathrm{TrunN}(1 - 0.78s + 0.22, 0.013; 0, 1) \text{ and } \alpha^{(2)}(s) = \mathrm{TrunN}(1 - 0.92s + 0.08, 0.029; 0, 1), \quad (9)$$

respectively, where $\mathrm{TrunN}(\mu, \sigma^2; 0, 1)$ is a truncated normal distribution with mean $\mu$ and variance $\sigma^2$ on interval $[0, 1]$. These ability profiles represent that both human workers and GenAI tools excel in easier skills but struggle with more challenging ones.

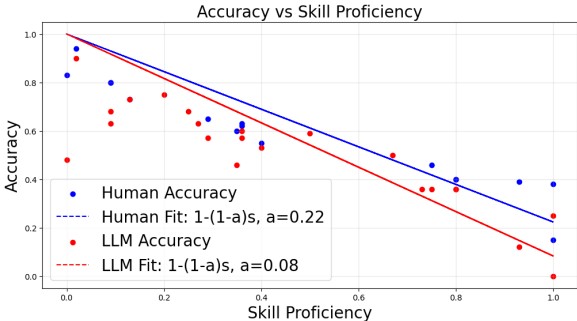

*Figure 11.* Accuracies of the average human rater and LLM v.s. skill proficiency for tasks in Big-bench Lite. We use linear functions to fit the plots. We fix the constant parameter $c = 1$ for ease of analysis such that the slope parameter $a$ can vary from 0 to 1. The variances for human and LLM are 0.013 and 0.029, respectively.

### E.3. Details for subskill division, task-skill dependency, and the choices of error functions

This section details how our framework can be adapted to the derived data from O*NET and Big-bench Lite.

**Deriving decision-level degree of skills.** To derive subskill numbers, we first need to know the decision-level degree of a skill. Suppose the decision-level subskill contributes $\lambda_j$-fraction and the action-level subskill contributes $(1 - \lambda_j)$-fraction for some $\lambda_j \in [0, 1]$. This quantization $\lambda_j$ depends on both the skills and the considered jobs.

To this end, we first use GPT-4o to obtain the description of the decision and action aspects of each skill. According to the descriptions, we also distinguish whether both the decision and action-level aspects can be evaluated separately. The prompt

*Table 3.* Accuracies of average human raters and the best LLM for 24 skills in BIG-bench Lite; information from Figure 4 of (bench authors, 2023). "NA" represents that the accuracy is unclear from the figure.

| Skill | Accruacy of average human rater | Accuracy of the best LLM |
|---|---|---|
| auto_debugging | 0.15 | NA |
| bbq_lite_json | 0.73 | 0.73 |
| code_line_description | 0.6 | 0.46 |
| conceptual_combinations | 0.83 | 0.48 |
| conlang_translation | NA | 0.5 |
| emoji_movie | 0.94 | 0.9 |
| formal_fallacies | 0.55 | 0.53 |
| hindu_knowledge | NA | 0.75 |
| known_unknowns | 0.8 | 0.68 |
| language_identification | NA | 0.36 |
| linguistics_puzzles | NA | NA |
| logic_grid_puzzle | 0.4 | 0.36 |
| logical_deduction | 0.4 | 0.36 |
| misconceptions_russian | NA | 0.68 |
| novel_concepts | 0.65 | 0.57 |
| operators | 0.46 | 0.36 |
| parsinlu_reading_comprehension | NA | 0 |
| play_dialog_same_or_different | NA | 0.63 |
| repeat_copy_logic | 0.39 | 0.12 |
| strange_stories | 0.8 | 0.63 |
| strategyqa | 0.62 | 0.6 |
| symbol_interpretation | 0.38 | 0.25 |
| vitamin_fact_verification | 0.63 | 0.57 |
| winowhy | NA | 0.59 |

is "# Table 2. Given the list of skills for the job of Computer programmers from O*NET, please provide the description of the decision-level and action-level aspects for each skill. Moreover, for each skill, determine which one of its decision and action aspects is more essential and whether both decision-level and action-level aspects can be evaluated separately. Output a LaTeX table in a box containing the above information." See Table 4 for a summary. For instance, the decision and action aspects of "Active Listening" are "Understanding Context" and "Engagement", respectively. It is an action-type skill and the decision and action aspects are difficult to be assessed separately. For such inseparable skills, we set $\lambda_j = 0$ for action ones and $\lambda_j = 1$ for decision ones. For the remaining separable skills, we use GPT-4o to derive a decision-level degree $\lambda_j$. This concludes the generation of the following vector $\lambda$ for decision-level degree:

$$\lambda = (0, 0, 1, 1, 1, .6, .7, .4, .4, 0, .3, 1, 1, .6, .7, .6, 0, .4) \in [0, 1]^n. \tag{10}$$

The prompt is "# Table 4. For each skill $j$ with "Separable = Y", please construct a decision-level degree $\lambda_j \in [0, 1]$ representing the decision-level degree while $1 - \lambda_j$ represents the action-level degree of skill $j$."

**Deriving subskill numbers from skill proficiency and decision-level degree.** First, we note that we can not measure the decision and action aspects of some skills separately. We take "Active Listening" as an example to illustrate how to determine subskill numbers for these skills. It is an action skill with $\lambda_j = 0$. Then, the difficulty of the decision-level subskill should be the easiest one, while the action-level subskill should be equal to the skill proficiency. This corresponds to $(s_{j1}, s_{j2}) = (0, s_j)$. The case of $\lambda_j = 1$ is symmetric. Thus, we provide the following assumption.

**Assumption E.1 (Extreme points for subskill allocation).** *We assume if $\lambda_j = 1$, $(s_{j1}, s_{j2}) = (s_j, 0)$; and if $\lambda_j = 0$, $(s_{j1}, s_{j2}) = (0, s_j)$.*

For the remaining skills $j$ that can be well divided into decision and action aspects, we have $\lambda_j \in (0, 1)$. To determine $s_{j1}$ and $s_{j2}$, we first analyze the desired properties of them. We take Programming as an example.

*Table 4.* Subskill descriptions. In the column of "Separable", "Y" represents that the decision-level and action-level aspects of skills can be tested separately; "N" represents that the skill can not be tested independently; and finally, "Decision"/"Action" represents that skills cannot be tested separately but can be tested independently, and the main aspect is decision/action, respectively. For "N" skills, a possibility is to use group-based assessments, e.g., assign a group project with clear dependencies among team members to test **Social Perceptiveness + Coordination**; or simulate a customer service scenario where candidates must listen to customer concerns and respond effectively to test **Speaking + Active Listening**.

| Skill id | Skill name | Decision | Action | Separable |
|---|---|---|---|---|
| 1 | Coordination | Planning Interactions - Identifying interdependencies | Adjusting Actions - Modifying behavior to align with others | N, action |
| 2 | Social Perceptiveness | Recognizing Cues - Understanding social signals | Response - Adjusting behavior based on social understanding | N, action |
| 3 | Mathematics | Conceptual Analysis - Choosing appropriate methods | Calculation - Executing mathematical computations | Decision |
| 4 | Time Management | Prioritization - Deciding task importance | Scheduling - Allocating time to tasks | N, decision |
| 5 | Monitoring | Identifying Key Indicators - Determining what to monitor | Observation - Actively tracking performance | Decision |
| 6 | Systems Analysis | System Design - Understanding how changes affect outcomes | Application - Using systems knowledge to modify systems | Y |
| 7 | Judgment and Decision Making | Weighing Options - Assessing risks and benefits | Execution - Choosing and enacting the best course | Y |
| 8 | Writing | Planning Content - Structuring and organizing ideas | Execution - Writing clearly and coherently | Y |
| 9 | Active Listening | Understanding Context - Interpreting information | Engagement - Showing attentiveness through responses | N, action |
| 10 | Speaking | Content Selection - Deciding what to convey | Delivery - Articulating information effectively | N, action |
| 11 | Quality Control Analysis | Standards Evaluation - Deciding quality benchmarks | Inspection - Physically testing or inspecting outcomes | Y |
| 12 | Reading Comprehension | Interpretation - Extracting key ideas from text | Application - Using information in a practical context | Decision |
| 13 | Systems Evaluation | Assessing Performance - Setting criteria for evaluation | Monitoring - Observing system function relative to criteria | N, decision |
| 14 | Operations Analysis | Determining Requirements - Identifying needs | Implementation - Designing solutions based on analysis | Y |
| 15 | Complex Problem Solving | Analyzing Options - Identifying potential solutions | Implementation - Applying solutions to problems | Y |
| 16 | Critical Thinking | Evaluating Alternatives - Comparing pros and cons | Logical Application - Applying chosen solution | Y |
| 17 | Active Learning | Identifying Relevance - Deciding useful information | Application - Using new information to solve tasks | Y |
| 18 | Programming | Designing Algorithms - Choosing the best approach | Writing Code - Implementing code in specific languages | Y |

- Fixing $\lambda_j$ and increasing $s_j$ (i.e., increasing the programming difficulty), we expect that the difficulties of both decision and action aspects increase, leading to an increase in $s_{j1}$ and $s_j2$.

- Fixing $s_j$ and increasing $\lambda_j$ (i.e., increasing the importance of decision-making in programming), we expect that $s_{j1}$ is closer to $s_j$. Specifically, when $\lambda_j = 1$, we have $s_{j1} = s_j$ by Assumption E.1. Moreover, we expect that $s_{j2}$ decreases since the requirement of action-level becomes easier. Symmetrically, as $\lambda_j$ decreases, we expect that $s_{j1}$ decreases and $s_{j2}$ is closer to $s_j$.

These desired properties motivate the following assumption.

**Assumption E.2 (Monotonicity for subskill allocation).** *We assume 1) $s_{j1}$ and $s_{j2}$ are monotonically increasingly as $s_j$; and 2) When $\lambda_j$ increases from 0 to 1, $s_{j1}$ is monotonically increasing from 0 to $s_j$ while $s_{j2}$ is monotonically decreasing from $s_j$ to 0.*

Finally, note that $s_{j1}$ and $s_{j2}$ are derived from skill proficiency $s_j$ and $\lambda_j$ only affects the allocation instead of the total skill difficulty. Thus, we would like a recovery of $s_j$ using $s_{j1}$ and $s_{j2}$. Observed from Assumption E.1, we may expect that $s_{j1} + s_{j2} = s_j$ holds. Accordingly, we have the following assumption.

**Assumption E.3 (Subskill complementarity assumption).** *We assume that $s_{j1} + s_{j2} = s_j$.*

Under Assumptions E.1-E.3, we conclude the following unified form of $s_{j1}$ and $s_{j2}$:

$$s_{j1} = \psi(\lambda_j)s_j \text{ and } s_{j2} = (1 - \psi(\lambda_j))s_j,$$

where $\psi(\cdot) : [0, 1] \to [0, 1]$ is a monotonically increasing function with $\psi(0) = 0$ and $\psi(1) = 1$. The easiest way is to select $\psi(\lambda) = \lambda$, which results in

$$s_1 = (0, 0, .45, .45, .45, .27, .322, .184, .184, 0, .15, .5, .52, .324, .385, .33, 0, .28) \in [0, 1]^n \text{ and}$$
$$s_2 = (.41, .43, 0, 0, 0, .18, .138, .276, .276, .48, .35, 0, 0, .216, .165, .22, .57, .42) \in [0, 1]^n. \tag{11}$$

This choice makes $s_{j1}$ and $s_{j2}$ proportional to $\lambda_j$. Other choices of $\psi$ include $\psi(\lambda) = \lambda^2$, $\psi(\lambda) = \frac{\lambda}{\lambda+1-(1-\lambda)e^{-\lambda}}$, and so on.

**Deriving subskill ability profiles.** We provide an approach to decompose skill ability profiles $\alpha$ to subskill ability profiles $\alpha_1$ and $\alpha_2$. When $\alpha(s) \sim \text{TrunN}(1 - (1 - a)s, \sigma^2; 0, 1)$ and the decision-level degree is $\lambda \in [0, 1]$, we set

$$\alpha_1(s) = \alpha_2(s) = \text{TrunN}(1 - (1 - a)s, \sigma^2/2; 0, 1).$$

This formula ensures that

$$1 - \alpha_1(s_{j1}) + 1 - \alpha_2(s_{j2}) = 2 - \text{TrunN}(1 - (1 - a)s_{j1}, \sigma^2/2; 0, 1) - \text{TrunN}(1 - (1 - a)s_{j2}, \sigma^2/2; 0, 1)$$
$$\approx \text{TrunN}((1 - a)(s_{j1} + s_{j2}), \sigma^2; 0, 1) \approx 1 - \text{TrunN}(1 - (1 - a)(s_{j1} + s_{j2}), \sigma^2; 0, 1) = 1 - \alpha(s_j),$$

where the last equation applies the property that $s_{j1} + s_{j2} = s_j$. This ensures that the distribution of $h(\zeta_{j1}, \zeta_{j2})$ is close to first draw $X \sim \alpha(s_j)$ and then outputs $1 - X$. Thus, we can (approximately) recover the skill ability profile via such subskill ability division by setting the skill success probability function $h(\zeta_1, \zeta_2) = \zeta_1 + \zeta_2$. Consequently, we have

$$\alpha_\ell^{(1)}(s) = \text{TrunN}(1 - 0.78s, 0.0065; 0, 1) \text{ and } \alpha_\ell^{(2)}(s) = \text{TrunN}(1 - 0.92s, 0.0145; 0, 1). \tag{12}$$

In conclusion, we provide an approach to divide the data on skills into subskill numbers and ability profiles.

**Details for deriving task-skill dependency.** Given the descriptions of tasks and skills for the job of "Computer Programmers", we use GPT-4o to generate the task-skill dependency $T_i$s; see Figure 12. The prompt is: "# Tables 1 and 2. Given a list of $m = 17$ tasks with their descriptions and a list of $n = 18$ skills with their descriptions in the job of Computer Programmers, please construct a subset $T_i \subseteq [n]$ for each task $i \in [m]$ that contains all skills $j$ associated to task $i$." The resulting task-skill dependency is: $T_1 = [6, 8, 9, 16, 18]$, $T_2 = [5, 7, 11, 16, 18]$, $T_3 = [5, 13, 14, 16, 18]$, $T_4 = [1, 4, 13, 18]$, $T_5 = [2, 7, 10, 17]$, $T_6 = [6, 11, 16, 18]$, $T_7 = [6, 8, 9, 18]$, $T_8 = [8, 11, 16, 18]$, $T_9 = [1, 2, 10, 17]$, $T_{10} = [5, 13, 14, 18]$, $T_{11} = [2, 8, 9, 10]$, $T_12 = [7, 13, 14, 18]$, $T_{13} = [1, 4, 7, 10]$, $T_{14} = [7, 8, 16, 18]$, $T_{15} = [6, 11, 16, 18]$, $T_{16} = [7, 10, 17, 18]$, and $T_{17} = [9, 16, 17, 18]$.

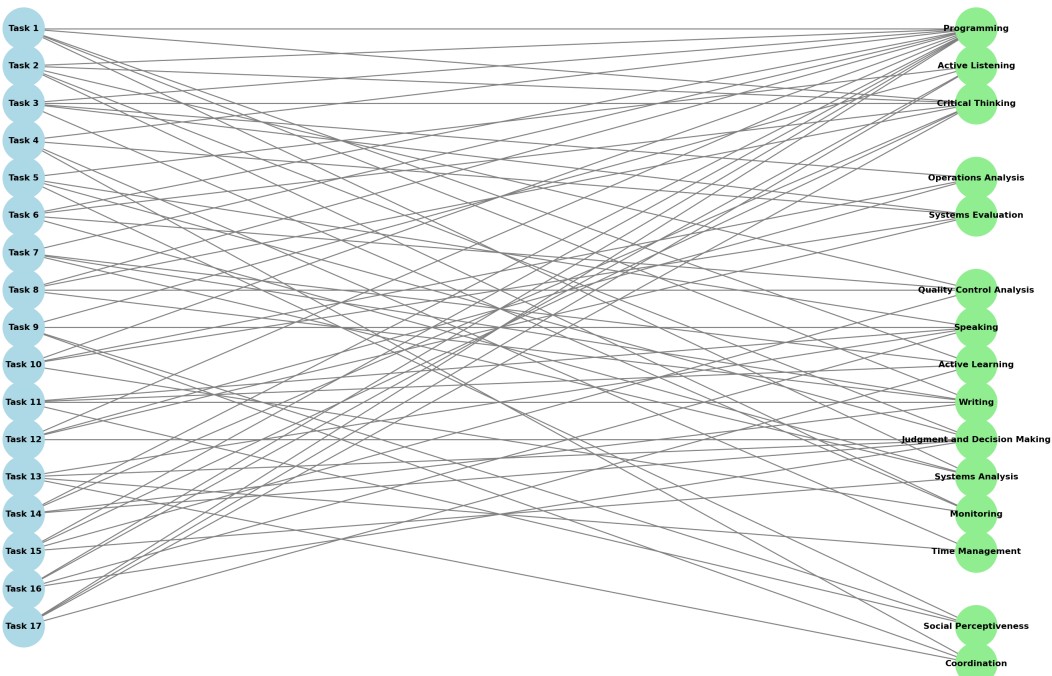

Task-Skill Relationship Graph (Adjusted for Readability)

*Figure 12.* Task-skill dependency graph for the Computer Programmers example. In this graph, $T_1 = [6, 8, 9, 16, 18]$, $T_2 = [5, 7, 11, 16, 18]$, $T_3 = [5, 13, 14, 16, 18]$, $T_4 = [1, 4, 13, 18]$, $T_5 = [2, 7, 10, 17]$, $T_6 = [6, 11, 16, 18]$, $T_7 = [6, 8, 9, 18]$, $T_8 = [8, 11, 16, 18]$, $T_9 = [1, 2, 10, 17]$, $T_{10} = [5, 13, 14, 18]$, $T_{11} = [2, 8, 9, 10]$, $T_12 = [7, 13, 14, 18]$, $T_{13} = [1, 4, 7, 10]$, $T_{14} = [7, 8, 16, 18]$, $T_{15} = [6, 11, 16, 18]$, $T_{16} = [7, 10, 17, 18]$, and $T_{17} = [9, 16, 17, 18]$.

**Choice of error functions.** As discussed above, we select the skill error function $h$ to be $h(\zeta_1, \zeta_2) := \zeta_1 + \zeta_2$ that takes realized subskill abilities $\zeta_1, \zeta_2$ as inputs and outputs a skill completion quality. This choice of $h$ aims to recover the derived skill ability function $\alpha$ from Big-bench Lite. Using the skill importance $w$, we select the task error function $g$ to be $g((h_j)_{j \in T_i}) := \frac{1}{\sum_{j \in T_i} w_j} \sum_{j \in T_i} w_j \cdot h_j$ that takes associated skill completion qualities of task $i$ as inputs and outputs a task completion quality. This choice of $g$ highlights the different importance of skills for the job. Finally, using the task importance $v$, we select the job error function $f$ to be $f(g_1, \ldots, g_m) := \frac{1}{\sum_{j \in T_i} v_i} \sum_{i \in [m]} v_i \cdot g_i$ that takes all task completion qualities as inputs and outputs a job completion quality. Combining with the task-skill dependency, we can compute the following function of job error rate composed by $h, g, f$: for any $\zeta \in [0, 1]^{2n}$,

$$
\begin{aligned}
\mathsf{Err}(\zeta) := \quad & 0.04(\zeta_{1,1} + \zeta_{1,2}) + 0.04(\zeta_{2,1} + \zeta_{2,2}) + 0.03(\zeta_{4,1} + \zeta_{4,2}) + 0.03(\zeta_{5,1} + \zeta_{5,2}) + 0.05(\zeta_{6,1} + \zeta_{6,2}) \\
& + 0.07(\zeta_{7,1} + \zeta_{7,2}) + 0.06(\zeta_{8,1} + \zeta_{8,2}) + 0.05(\zeta_{9,1} + \zeta_{9,2}) + 0.06(\zeta_{10,1} + \zeta_{10,2}) + 0.05(\zeta_{11,1} + \zeta_{11,2}) \\
& + 0.05(\zeta_{13,1} + \zeta_{13,2}) + 0.04(\zeta_{14,1} + \zeta_{14,2}) + 0.11(\zeta_{16,1} + \zeta_{16,2}) + 0.06(\zeta_{17,1} + \zeta_{17,2}) \\
& + 0.26(\zeta_{18,1} + \zeta_{18,2}).
\end{aligned}
\tag{13}
$$

Overall, we show how to derive all the data for using our framework. We can simulate that the job success probabilities of $W_1$ and $W_2$ are $P_1 = 0.55$ and $P_2 = 0.00$, respectively. We also provide a flow chart to summarize this procedure; see Figure 13. We remark that we can compute $P_1$ and $P_2$ even without subskill division, i.e., only using data including skill proficiencies as in Equation (8), skill ability profile as in Equation (9), and the function of job error rate as in Equation (13). For $\tau = 0.45$, we obtain that $P_1 = 0.84$ and $P_2 = 0.00$. The value of $P_1$ is different but not too far from that computed using the subskill division, which is convincing of the reasonability of our subskill division approaches.

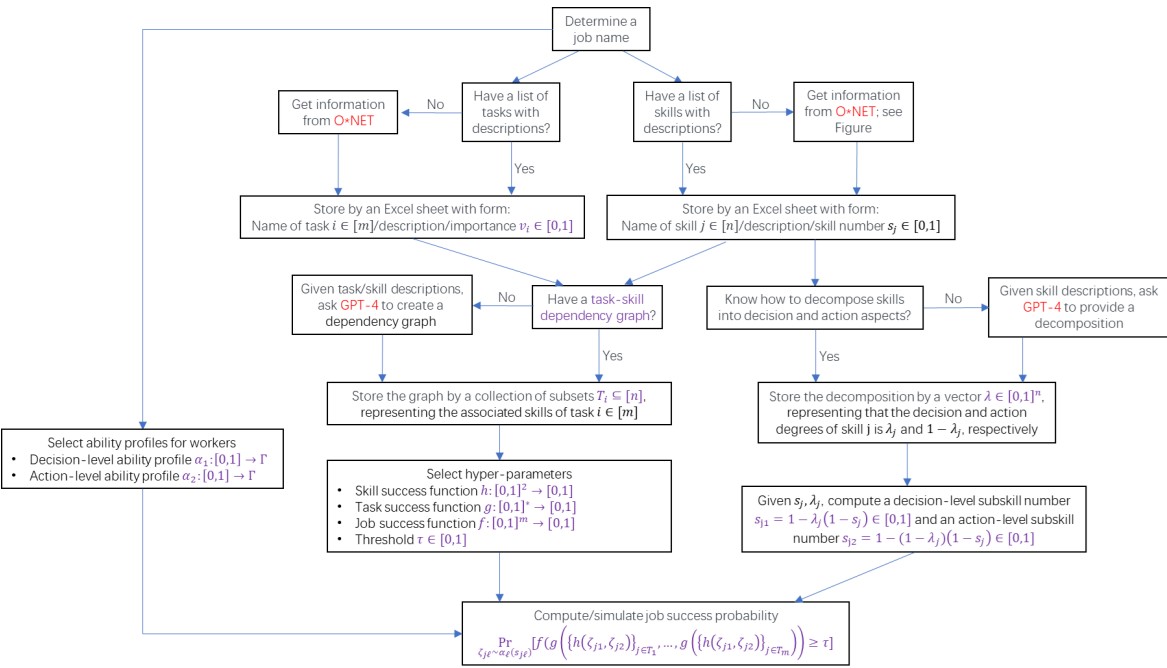

*Figure 13.* A flow chart for the Computer Programmers example that illustrates how to use our framework to assess job-worker fit.

### E.4. Robustness across alternative modeling choices

Besides the use of the derived job and worker data in Section 4, we also do simulations with alternative modeling choices to validate the robustness of our findings.

**Alternative error functions.** We replace the job/task error aggregation functions $g$ and $f$ with max to simulate more fragile task environments; see Figures 14 and 15. The main patterns remain consistent with those for average error functions, though line-crossings disappear due to monotonicity in the max-based error aggregation.

**Alternative ability distributions.** We substitute truncated normals with uniform noise in ability profiles (Figures 16 and 17), verifying that our key findings hold across distributions.

**Robustness to task-skill graph variations.** We randomly modify 5 edges in the task-skill dependency graph (Figures 18 and 19). Despite these changes, the phase transition behavior and heatmaps remain stable.

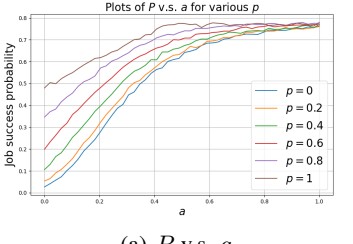
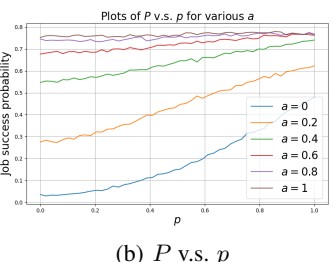
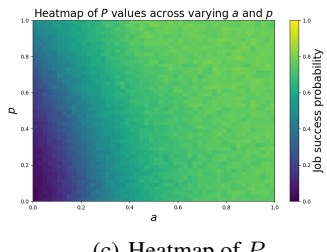

(a) $P$ v.s. $a$          (b) $P$ v.s. $p$          (c) Heatmap of $P$

*Figure 14.* Plots illustrating the relationship between the success probability $P(\alpha_1, \alpha_2, h, g, f, \tau)$ and the ability parameter $a$ and dependency parameter $p$ for the Computer Programmers example with default settings of $(\sigma, \tau) = (0.08, 0.6)$, replacing the error functions $g, f$ from weighted average in Section 4 to max. Note that we increase $\tau$ from 0.45 (for weighted average) to 0.6 (for max), since the resulting error rate of max is higher. The job structure and worker ability profiles follow the same design as Figure 3 in our main paper, demonstrating the robustness of our empirical results for the job error rate function JER.

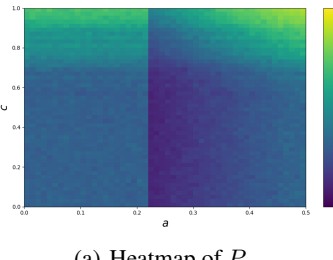
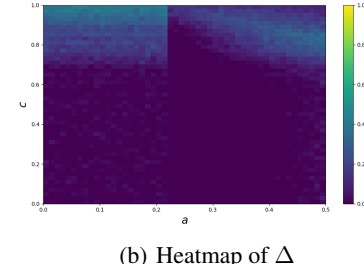

(a) Heatmap of $P_{merge}$          (b) Heatmap of $\Delta$

*Figure 15.* Heatmaps of job success probability $P_{merge}$ and the probability gain $\Delta = P_{merge} - \max\{P_1, P_2\}$ by merging two workers for different ranges of $(a, c)$ for the Computer Programmers example with default settings of $\tau = 0.6$, replacing the error functions $g, f$ from weighted average in Section 4 to max. Note that we increase $\tau$ from 0.45 (for weighted average) to 0.6 (for max), since the resulting error rate of max is higher. The job structure and worker ability profiles follow the same design as Figure 4 in our main paper, demonstrating the robustness of our empirical results for the job error rate function JER.

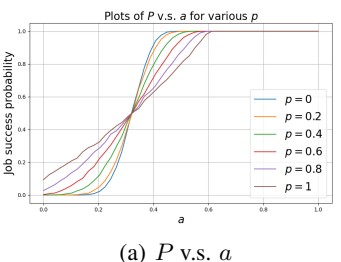
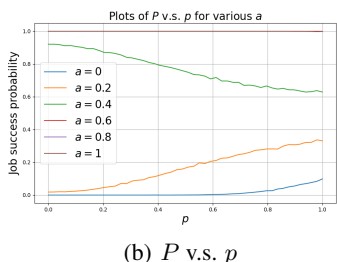
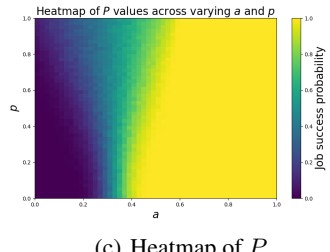

(a) $P$ v.s. $a$          (b) $P$ v.s. $p$          (c) Heatmap of $P$

*Figure 16.* Plots illustrating the relationship between the success probability $P(\alpha_1, \alpha_2, h, g, f, \tau)$ and the ability parameter $a$ and dependency parameter $p$ for the Computer Programmers example with default settings of $(\sigma, \tau) = (0.2, 0.4)$, replacing the truncated normal noise in Section 4 with the uniform noise. Setting $\sigma = 0.2$ ensures that the variance of the uniform distribution matches that of the truncated normal distribution. The job structure and the job error rate function JER follow the same design as Figure 3 in our main paper, demonstrating the robustness of our empirical results for worker ability profiles.

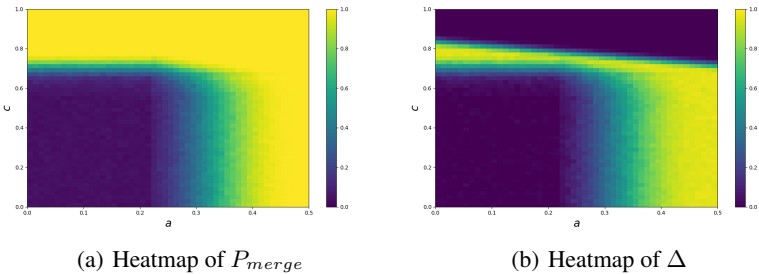

(a) Heatmap of $P_{merge}$        (b) Heatmap of $\Delta$

*Figure 17.* Heatmaps of job success probability $P_{merge}$ and the probability gain $\Delta = P_{merge} - \max\{P_1, P_2\}$ by merging two workers for different ranges of $(a, c)$ for the Computer Programmers example with default settings of $(\sigma_1, \sigma_2, \tau) = (0.2, 0.29, 0.4)$, replacing the truncated normal noise in Section 4 with the uniform noise. The variance parameters $\sigma_1$ and $\sigma_2$ are chosen so that the variance of the uniform noise for both workers aligns with that of the truncated normal distribution. The job structure and the job error rate function JER follow the same design as Figure 4 in our main paper, demonstrating the robustness of our empirical results for worker ability profiles.

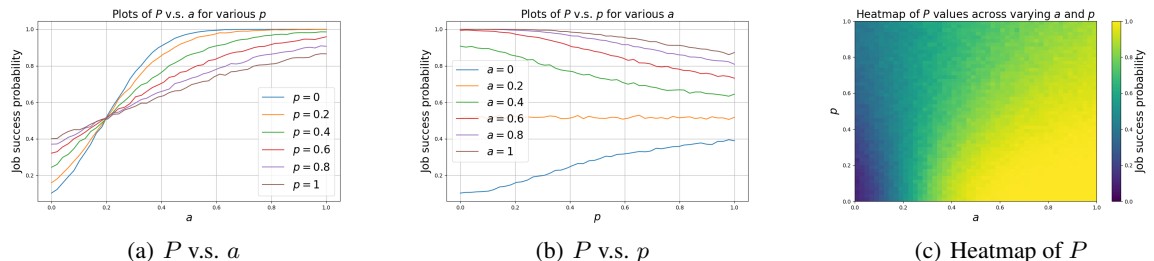

(a) $P$ v.s. $a$        (b) $P$ v.s. $p$        (c) Heatmap of $P$

*Figure 18.* Plots illustrating the relationship between the success probability $P(\alpha_1, \alpha_2, h, g, f, \tau)$ and the ability parameter $a$ and dependency parameter $p$ for the Computer Programmers example with default settings of $(\sigma, \tau) = (0.08, 0.45)$, randomly shifting five edges in the task-skill dependency graph. The worker ability profiles and the job error rate function JER follow the same design as Figure 3 in our main paper, demonstrating the robustness of our empirical results for job structure.

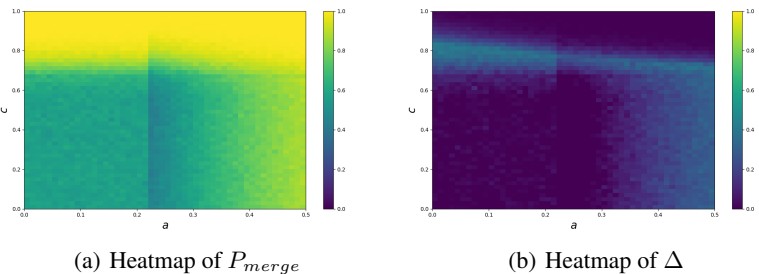

(a) Heatmap of $P_{merge}$        (b) Heatmap of $\Delta$

*Figure 19.* Heatmaps of job success probability $P_{merge}$ and the probability gain $\Delta = P_{merge} - \max\{P_1, P_2\}$ by merging two workers for different ranges of $(a, c)$ for the Computer Programmers example with default settings of $\tau = 0.45$, randomly shifting five edges in the task-skill dependency graph. The worker ability profiles and the job error rate function JER follow the same design as Figure 4 in our main paper, demonstrating the robustness of our empirical results for job structure.

