# OpenReview forum: "A Mathematical Framework for AI-Human Integration in Work"
_ICML.cc/2025/Conference — ICML 2025 poster_

### Official Review · Reviewer_1zaz · 2025-03-13

**Overall Recommendation:** 4

**Summary:**

This paper develops a model of job success probability by viewing jobs as a composition of tasks that need to be accomplished, and workers supply skills that affect the probability tasks as successfully completed. The authors then calibrate the model using the O*NET database's skill descriptions associated with computer programming. The authors calibrate the skill of workers and the skill of GenAI tools in their model to examine whether merging the human + the GenAI tool can lead to higher job success probabilities than either working in isolation. I provide a more detailed description of the theoretical model and experimental results before presenting my comments below.

**Claims And Evidence:**

Please see my discussion of theoretical claims and experimental designs.

**Essential References Not Discussed:**

The paper provides an thorough review of empirical research studying the impacts of generative AI on worker productivity. There are no glaring omissions to me from this literature. In particular, I am not aware of existing paper that attempts to write a stylized model to understand of (i) jobs as a composition of tasks to be completed, and (ii) skills leading to imperfect completion rates of underlying tasks.

**Experimental Designs Or Analyses:**

To illustrate the model, the authors derives data on tasks and skills for computer programmers using O*NET. O*NET describes a computer programmer as consisting of 18 skills and 17 tasks with associated with proficiency levels for each skill. The authors prompt GPT-4o to provide the relationship between skills and tasks, the division of skill proficiencies into action and decision skills, and prompt GPT-4o using big-bench lite to construct skill profiles for an average person and an LLM.

(1) Prompting GPT-4o plays a key role in the empirical results. It would have been useful for the authors to describe this more explicitly in the main text -- in particular, the GPT-4o outputs are used to build the task-skill dependency network; GPT-4o outputs are used to divide proficiencies into decision and action skills; and GPT-4o outputs are used to construct the proficiencies of LLMs and human rates based on big-bench lite.  I am not sure how seriously to take this except as a way to construct some numbers used for the calibration of the model.

(2) The authors pick particular parametrizations of the skill profiles based on truncated normals -- how were these chosen? Why this specific choice and not the uniform alternative? It would be valuable to see the sensitivity of the calibration results to that specific modelling choice here.

(3) Related to my earlier comment about the generality of the choice of skill, task and job functions, it would be interesting to see alternative variations on the choice of the JER function -- the authors choose a rather simple function that is the weighted average of the skills. What if you instead took the max error associated with each skill in a task and then max error across tasks? How would the comparisons change?

**Methods And Evaluation Criteria:**

Please see my discussion of theoretical claims and experimental designs.

**Other Comments Or Suggestions:**

As a small comment, between Section 2 and Section 3, the authors appear to switch notation for the average skill/ability. In Section 2, it is introduced as $E(s)$ and in Section 3 it is denoted by $\mu$. The paper would be easier to digest if the authors used consistent notation throughout.

**Other Strengths And Weaknesses:**

Please see my earlier comments.

**Questions For Authors:**

Please see my previous comments on the theoretical claims and the experiment.

**Relation To Broader Scientific Literature:**

As discussed by the authors in their introduction, a large and highly active literature studies the productivity effects of generative AI across a wide variety of settings. But much of this work lacks a clear theoretical framework for understanding when/how generative AI tools affect productivity. This paper aims to provide such a framework by viewing jobs as a collection of tasks that need to be completed and workers as having skills that affect their probabilities of successfully completing tasks. I find this to be a valuable exercise and a potentially important contribution to this active empirical literature. At the same time, as I discussed above, the model is simultaneously specific yet opaque. Moreover, there results provided are obvious -- Theorem 3.2 has a lot of leg work involved to show that increasing the average skill of a worker leads to an increase in the job success probability; and Theorem 3.3 has a lot of leg work involved to show that combining two workers that are more skilled along different dimensions leads to an increase in the job success probability. While the authors motivated the framework using these empirical findings, I struggled to link this model back to those findings.

**Theoretical Claims:**

The authors theoretical model has the following key components:
* A job consists of $m$ tasks and each task $T_i$ depends on $n$ skills. Each skill $s$ has two components: decision-level and action-level sub-skills.
* A worker in the model is associated with two ability profiles $\alpha_1(s), \alpha_2(s)$ that summarizes their ability across the two subskills for each skill. The ability profiles are governed by a probability distribution that is summarized by two parameters: (i) the average ability for the skill, and (ii) the noise for the skill.
* For each skill $j \in [n]$, the worker's skill error function is given by the random variable $h(\zeta_{j1}, \zeta_{j2})$ where $z_{j,l} = 1 - X_{jl}$ for $X_{jl}$ sampled according to the skill profile $\alpha_{l}(s_{jl})$. The task error then aggregates the skill errors with $g(h(\zeta_{11}, \zeta_{12}), ..., h(\zeta_{n1}, \zeta_{n2}))$. Finally, the job error aggregates the task errors $f \colon [0,1]^m \rightarrow 1$. Consequently, task success and ultimate job success are random variables due to the randomness in the worker's skill errors.
* The authors' main object of interest is the job success probability which is the probability the job error is less than some threshold $\tau$. (Equation 1)
* The authors then provide two results: (1) the authors show that the job success probability is increasing in the average skill of a worker, (2)  there can be gains in the job success probability by merging two workers together.

One of the strengths of the paper is the model's generality -- in particular, the setup can accommodate a wide variety of choices of skill errors, task errors, and job errors. Akin to the authors example of the max, you could carry this forward and imagine an "O-ring" (Kremer, 1993) style model which would involve: skill errors as being either zero or one, tasks $g() = 1$ only be completed if all skills are successfully completed  $h() = 1$ and jobs $f() = 1$ only being completed if all tasks are completed.

One of the weaknesses is that while the model is general, the results are relatively weak and unsurprising. Consequently, it is not clear why this model helps me reason through the suite of empirical studies that analyze the productivity effects of generative AI. The authors rely heavily on returning to stylized versions of the model to build intuition (e.g., linear ability functions) but I did not find that enlightening. I would have found it immensely valuable for the authors to either (i) pick a particular result (e.g., Brynjolfsson et al.), or (ii) recurring finding across empirical analyses (e.g., the finding that the introduction of GenAI tool leads to a compression in the productivity distribution across workers) and discuss whether the model has anything to say about those results.

Another weakness of the paper is that I struggled to map the model into any concrete job. Take the example of computer programmers that the authors study in the experiment. If I think about the model as describing one worker, it describes a worker completing many jobs and describes the fraction of jobs that the worker successfully completes. So, for a computer programmer, a job actually corresponds to a specific programming project and we would be describing the success rate of the computer programmer across many such jobs? In this view, the randomness in the skill successes arises from randomness in the worker's skill across such tasks (maybe some days I am tired and other days I have a lot of coffee?). I can see how this works for a job like programmer where there is a somewhat discrete output being produced, but for other jobs it is not obvious this applies.

---

> ### Author Rebuttal · Authors · 2025-04-01
>
> We thank you for your thoughtful, detailed, and insightful feedback. In response, we added new theoretical and empirical analyses that sharpen our results, test their robustness across modeling choices, and highlight connections to real-world phenomena such as productivity compression.
>
> Please see this PDF for new figures: https://acrobat.adobe.com/id/urn:aaid:sc:eu:fef7d6b2-24f6-4a59-9386-fdf09456ed99.
>
> > *"Pick a particular empirical result (e.g., Brynjolfsson et al.)... and discuss whether the model explains it."*
>
> Thank you for the suggestion. We show that our model formally supports the **productivity compression** effect observed in Brynjolfsson et al. (2023). Consider two workers from the same ability family (e.g., constant, linear, polynomial), with equal decision-level ability and noise: a low-skilled worker $W_1$ with action-level ability parameter $a_1$ and a high-skilled worker $W_2$ with action-level ability parameter $a_2>a_1$. Let $P_1,P_2$ be their success probabilities before merging with GenAI
> (whose decision-level ability is always weaker than $W_\ell$ and action-level ability is in the same family with $W_1$ and $W_2$), and $P'_1, P'_2$ after merging. We define productivity compression as: $\mathrm{PC}=(P_2-P_1)-(P'_2-P'_1)$.
>
> **Theoretical insight:** A corollary of Theorem 3.2 shows that if $a_{AI}>a_1$, then $W_1$’s merged gain $\Delta_1=P'_1-P_1$ can be large-up to $1-2\theta$ for some small $\theta$—while $W_2$'s gain $\Delta_2\approx 0$. Thus, $\mathrm{PC} \approx 1-2\theta$.
>
> **Empirical results:** Our experiments (Figure 6 in the PDF) show that even when the GenAI ability profile differs from that of the worker, compression increases with the skill gap. For $a_1=0.1$, $a_2=0.8$, compression reaches 0.8, closely matching empirical findings from Brynjolfsson et al.
>
> To our knowledge, this is one of the first formal explanations of the compression effect under realistic assumptions.
>
> > *"..uniform alternative?"*
>
> We chose truncated normal distributions to approximate ability variability observed in empirical benchmarks (e.g., Figure App.9 in Bench authors, 2023). That said, we now include experiments with uniform noise distributions (Figures 3 and 4 in the PDF) and find qualitatively similar results.
>
> > *"What if you instead took the max error..."*
>
> This excellent suggestion led us to study max-based aggregation, where job success probability is given by $P = \prod_{j} \Pr[h_j \leq \tau]$. We conducted both theoretical and empirical analyses:
>
> - **Sharper Phase Transition:** With identical skills, the transition window scales as $\frac{1}{n}$, sharper than the average-based case $\frac{1}{\sqrt{n}}$.
>
> - **Negative Example:** We identify a natural setting where no phase transition occurs. With one hard skill (e.g., 0.1) and others easy (e.g., 0.8), success is dominated by the hard one: $P \approx \Pr[h_1 \leq \tau]$, yielding a smooth, non-abrupt transition.
>
> - **Empirical Results:** Experiments (Figures 1 and 2 in the PDF) confirm these effects. Notably, line-crossing in average-based settings disappears under max aggregation due to its monotonicity.
>
> We will include these results in the final version.
>
> > *"struggled to map the model into any concrete job..."*
>
> Our model is intended to be flexible. For project-based roles (e.g., programming), a job may represent a single project composed of subtasks. For ongoing roles (e.g., teaching or customer service), a job could aggregate performance over a time period. The randomness in subskill outcomes captures intra-personal variability (e.g., fatigue) and task-level heterogeneity. While stylized, we believe this abstraction enables us to reason about human-AI collaboration in both discrete and continuous work environments.
>
> > *"Prompting GPT-4o plays a key role..."*
>
> We will clarify the role of GPT-4o in the main text and include API-based prompting code in the final version to ensure consistency and reproducibility.
>
> > *“Results are relatively weak and unsurprising.”*
>
> While some results may seem intuitive, our theorems precisely characterize when phase transitions in job success occur and quantify non-trivial merging gains—analytically nontrivial and not evident without formal modeling. Our extensions further show when phase transitions disappear (e.g., max-based errors, strong skill dependencies), underscoring the value of our framework.
>
> > *“Rely heavily on stylized versions of the model (e.g., linear ability functions)...”*
>
> We use linear ability functions as an interpretable baseline aligned with prior empirical work (e.g., Brynjolfsson et al., 2023). This choice illustrates key effects clearly and enables tractable analysis. In Appendix C.3, we extend our results to polynomial ability functions, and our experiments confirm that the core insights persist under more complex profiles.
>
> We thank you again for your thoughtful engagement, which has helped us clarify, extend, and strengthen the contributions of our work.

---

> > ### Comment · Reviewer_1zaz · 2025-04-07
> >
> > I thank the authors for thoughtfully engaging with my review. I was originally positive about the paper, but I am encouraged by the authors new discussion about how the model can explain the ``compression effect'' of GenAI tools documented in existing empirical evaluations. I will revise my score upwards, and I would strongly encourage the authors to further emphasize how the model can make sense of empirical findings on the productivity effects of GenAI tools.

---

### Official Review · Reviewer_ChBu · 2025-03-15

**Overall Recommendation:** 3

**Summary:**

This paper presents a mathematical framework for modeling jobs, workers, and worker-job fit, focusing on subskill decomposition into decision-level and action-level tasks to highlight the distinct strengths of humans and AI. The study examines how variations in subskill abilities affect job success and identifies conditions under which collaborative skill division leads to superior performance compared to relying on a single worker. The framework's effectiveness is validated using O*NET and Big-Bench Lite datasets, demonstrating its real-world applicability. The results emphasize that Generative AI (GenAI) is best suited to complement human workers' skills rather than replacing them.

**Claims And Evidence:**

Yes, the claims made in the submission are well-supported. For example, the authors argue that conflating reasoning skills with action skills can lead to misattributions of success or failure, resulting in biased or incomplete assessments. To address this, they provide a detailed framework for decomposing skills into two distinct subskill types: decision-level skills (problem-solving) and action-level skills (solution execution), with skill difficulty quantified on a 0-1 scale.

**Essential References Not Discussed:**

I am not familiar with workforce optimization, so I cannot fully assess the essence of the related work.

**Experimental Designs Or Analyses:**

The approach appears comprehensive and sound. The authors define a job-success probability metric that integrates error rates across skills and tasks to assess overall performance. This metric provides a structured and quantitative evaluation of human-AI collaboration effectiveness, ensuring a more holistic assessment of task execution.

**Methods And Evaluation Criteria:**

Yes, the proposed methods and evaluation criteria are well-aligned with the problem being addressed. The paper introduces a mathematical framework that systematically decomposes skills into decision-level and action-level subskills, which is a logical and structured approach to analyzing worker-job fit in the context of human-AI collaboration.

For evaluation, the use of O*NET and Big-Bench Lite datasets provides a real-world grounding for their framework, ensuring that the findings are not purely theoretical. These datasets contain job-related skill data and AI task benchmarks, making them appropriate for assessing the practicality of skill decomposition.

**Other Comments Or Suggestions:**

The study uses O*NET and Big-Bench Lite, but it would be useful to discuss potential biases or limitations in these datasets.
Are there task distributions that may favor either AI or human workers?

Beyond theoretical modeling, empirical user studies involving real-world AI-human collaboration could strengthen the paper’s conclusions.

**Other Strengths And Weaknesses:**

A potential limitation is whether the chosen datasets fully capture the complexity of skill attribution in dynamic work environments. Additional experiments with task-specific benchmarks or real-world AI-assisted work scenarios could further validate the robustness of the proposed approach.

**Questions For Authors:**

Please refer to Section of Other Comments Or Suggestions.

**Relation To Broader Scientific Literature:**

By refining how AI-human performance is measured, this work contributes to future AI deployment strategies, helping design collaborative work environments where AI complements human abilities rather than replacing them. The proposed methodologies provide insights into the effective allocation of human and AI resources, ultimately improving job success probability and fostering productive human-AI collaboration.

**Theoretical Claims:**

I am not very familiar with the topic of assessing job accuracy of AI and humans, so I cannot fully judge whether all the theoretical components are meaningful. However, I have reviewed the mathematical symbol definitions, and they are clearly defined and well-presented.

---

> ### Author Rebuttal · Authors · 2025-04-01
>
> We thank you for your detailed and encouraging review. We are especially grateful for your recognition of our framework’s real-world applicability and for your thoughtful suggestions regarding dataset limitations and empirical grounding, which have directly shaped our additional experiments. Please see this PDF for new figures (https://acrobat.adobe.com/id/urn:aaid:sc:eu:fef7d6b2-24f6-4a59-9386-fdf09456ed99).
>
> > *"A potential limitation is whether the chosen datasets fully capture the complexity of skill attribution in dynamic work environments. Additional experiments with task-specific benchmarks or real-world AI-assisted work scenarios could further validate the robustness of the proposed approach."*
>
> We agree that additional empirical results could further ground our mathematical model and its implications. As one of the first formal frameworks for understanding task-level human-AI collaboration, our empirical results—based on O\*NET and Big-Bench Lite—serve primarily as calibration tools to illustrate how the model can be instantiated and to demonstrate that key theoretical insights remain valid in practice. While we recognize the limitations of these datasets—O\*NET reflects static, survey-based data, and LLM-based estimates may introduce bias—we have conducted several new empirical analyses (see accompanying PDF) to test the robustness of our findings across alternative modeling choices. We will describe these limitations clearly and include the new results in the final version. We also agree that task-specific benchmarks (e.g., HumanEval for programming, customer support transcripts) could further strengthen empirical grounding and plan to explore such extensions in future work.
>
> - **Alternative Error Functions:**
>   We replace the job/task error aggregation functions $g$ and $f$ with $\max$ (suggested by Reviewer 1zaz), to simulate more fragile task environments. The main patterns remain consistent (Figures 1 and 2 in the PDF linked above), though line-crossings disappear due to monotonicity in the max-based error aggregation.
>
> - **Alternative Ability Distributions:**
>   We substitute truncated normals with uniform noise in ability profiles (Figures 3 and 4 in the PDF linked above), verifying that our key findings hold across distributions.
>
> - **Robustness to Task-Skill Graph Variations:**
>   We randomly modify 5 edges in the task-skill dependency graph (Figures 7 and 8 in the PDF linked above). Despite these changes, the phase transition behavior and heatmaps remain stable.
>
>
> > *"It would be useful to discuss potential biases or limitations in these datasets... are there task distributions favoring humans or AI?"*
>
> We will explicitly discuss biases in O\*NET and Big-Bench Lite datasets regarding task distributions potentially favoring humans or AI. We summarize key observations below:
>
> - **Tasks Favoring Humans:**
>   Tasks that are context-rich and require nuanced judgment, creativity, or interpersonal skills (e.g., strategic planning, ethical decisions) are better captured by human-centered data like O\*NET.
>
> - **Tasks Favoring AI:**
>   Structured, repetitive tasks (e.g., basic arithmetic, data classification) are common in benchmarks like Big-Bench Lite, which may overrepresent tasks where AI excels.
>
> These differences suggest that while O\*NET may underrepresent emerging digital tasks, Big-Bench Lite might favor tasks with clear, rule-based responses.
>
> > *"Beyond theoretical modeling, empirical user studies involving real-world AI-human collaboration could strengthen the paper’s conclusions."*
>
> We agree this would be valuable, though it is outside the scope of this paper. Our primary goal is to provide a rigorous theoretical framework with proof-of-concept empirical validation. We believe our framework offers a foundation for future empirical and behavioral studies on AI-assisted work. As noted, several of our new experiments aim to move further in that direction.
>
> We thank you again for your thoughtful comments, which have significantly improved our work's clarity and empirical reach. We hope our additions meaningfully address the concerns raised.

---

> > ### Comment · Reviewer_ChBu · 2025-04-04
> >
> > Thank the authors for providing detailed explanations in this rebuttal.

---

### Official Review · Reviewer_aCoN · 2025-03-15

**Overall Recommendation:** 3

**Summary:**

This paper models human-AI collaboration in jobs. In particular, it models jobs as being composed of multiple different subtasks, each of which involve different skills. The ability of different agents is noisy and ordered (e.g. the same agent can’t perform worse on easier subtasks on average than they do on easier subtasks). This paper studies multiple effects within this model, such as how the probability of task success varies with average ability and the effect of “merging” multiple workers (e.g. combining their relative strengths).

**Claims And Evidence:**

I appreciated that the paper took an especially nuanced view on how humans may integrate algorithmic tools into their workflow.The paper includes a rigorous theoretical analysis of the problem, as well as an empirical analysis.

**Essential References Not Discussed:**

The idea of modeling jobs as given by multiple subtasks has been studied extensively in prior work, especially since some core parts of this paper could also model human-human integration (e.g. merging multiple jobs into one).  For example, this lecture https://economics.mit.edu/sites/default/files/2024-09/Autor-Schumpeter-Expertise-20240829-handout.pdf, and related papers https://www.nber.org/system/files/working_papers/w32140/w32140.pdf studies similar issues.

**Experimental Designs Or Analyses:**

N/A

**Methods And Evaluation Criteria:**

This paper has a fairly involved theoretical model of tasks - my suspicion is that most of the core results would generalize to other models, but given that the results are relatively intuitive, it makes me wonder whether another (or a simpler model) would have sufficed.

One core concept that this paper explores is that of “merging workers”, which in this context likely means the benefits from partial automation of jobs. Given the model where humans and AI tools may have complementary skills sets (e.g. in decision-level or action-level skills), it seems natural that there would be benefits in merging workers. However, empirical work has shown that humans are sometimes unable to know when algorithmic tools are more or less accurate (e.g. https://arxiv.org/abs/2406.01382). It would have been useful to see more discussion of how these results change when workers are imperfectly “merged”.

**Other Comments Or Suggestions:**

N/A

**Other Strengths And Weaknesses:**

This is more a stylistic point, but the paper is very densely written, where the main body of the paper often includes more technical details than is necessary, which may make it difficult for the reader to follow the high-level story. Making the main body be written at a more high level (removing more technical discussion to the appendix) would be helpful.

**Questions For Authors:**

N/A

**Relation To Broader Scientific Literature:**

See below

**Theoretical Claims:**

N/A

---

> ### Author Rebuttal · Authors · 2025-04-01
>
> Thank you for your thoughtful and constructive feedback. We especially appreciate your recognition of our modeling approach and the suggestion regarding imperfect merging, which we have now incorporated. Please see this PDF for new figures (https://acrobat.adobe.com/id/urn:aaid:sc:eu:fef7d6b2-24f6-4a59-9386-fdf09456ed99).
>
> > "...when workers are imperfectly 'merged'."
>
> Thank you for this suggestion and the reference. We extend our merging analysis by introducing a trust parameter $\lambda$ to the merging experiment in Section 4 (Line 358 right column - 439 left column), which models imperfect merging by letting the estimated ability $\hat{c}=\lambda c$ deviate from the true action-level ability $c$ of worker $W_2$. We then assign action-level subskills to $W_2$ when its scaled ability, $\lambda c$, exceeds $W_1$'s ability (i.e., $1 - 0.78s_{j2}\le \lambda c$, extending the setting in Lines 414–415, left column), even though $W_2$ completes skills at level $c$. We analyze the probability gain $\Delta=P_{\text{merge}}-\max\\{P_1,P_2\\}$ across different values of $c$ and $\lambda$ (see Figure 5 in the PDF linked above).
>
> We find that even modest errors in $\lambda$ can sharply reduce $\Delta$. For example, when $\lambda=1.14$ and $c=0.2$, the probability gain becomes $\Delta=-0.2$, indicating that merging reduces job success. This illustrates the critical importance of accurate ability estimation, and complements the findings in Section E.2 on belief-driven merging.
>
> We will create a separate subsection in Appendix E to present these findings clearly.
>
> > "...wonder whether another (or a simpler model) would have sufficed."
>
> Our model combines a task-skill graph, subskill-level abilities, and error rate functions to estimate job success (Section 2). We clarify three key points:
>
> - **Generality:**
>   The model allows flexible task-skill dependencies, ability profiles, and error functions. Our framework includes two variants—noise in abilities and subskill division—to capture variation in worker performance. It can be adapted to simpler models while retaining its predictive power.
>
> - **Simplicity vs. expressiveness:**
>   A special case with a single task, noise-free abilities, and no subskill split yields a binary job success probability (0 or 1) and misses correlations across skills. Without subskill division, it is difficult to analyze how merging a human and GenAI tool—each excelling in different subskills—affects performance.
>
> - **Non-triviality of results:**
>   While some conclusions may seem intuitive, formally establishing phase transitions in job success requires careful analysis, as illustrated by the following examples:
>
>   - **Dependent skills:** By introducing a dependency parameter $p \in [0,1]$, subskill errors are drawn from a shared latent status with probability $p$ and independently otherwise. The phase transition window becomes $\gamma_1=\frac{L\sqrt{\mathrm{sg}(A_1)\cdot\ln\frac{1}{\theta}}}{\mathrm{Infg}(A_1)\cdot\sqrt{1-p}}$, which vanishes as $p \to 1$, showing that stronger dependencies smooth out abrupt transitions.
>   - **Max aggregation:**
>     We show that Theorem 3.1 may not yield a phase transition when using the max operator for error rate functions $g,f$. Consider a job where one critical task is very difficult (e.g., skill difficulty 0.1) while all other tasks are relatively easy (e.g., skill difficulties 0.8). Since the job error rate is defined as $\max_{j\in[n]}h_j$, the overall error is dominated by the hard task. In this case, the job success probability is essentially $P=\Pr[h_1\leq\tau]$, making it a smooth function of the ability on that single, critical task—thus, no sharp phase transition occurs. Notably, the Lipschitz constant here is significantly higher (e.g., $L=1$) compared to the average-case model (e.g., $L=1/(2n)$ in the linear case), which prevents an abrupt transition.
>
> These findings show that richer modeling is necessary to rigorously capture how noise, dependencies, and aggregation affect outcomes like merging. We will include these as theorems in the appendix.
>
> > "Modeling jobs as multiple subtasks has been studied..."
>
> Thank you for this pointer. We will cite related works (e.g., Autor and Thompson) and clarify that our contribution lies in offering a formal, ML-grounded framework for modeling skill decomposition and human-AI complementarity. While inspired by similar questions, our analysis tackles fine-grained questions around merging, noise, and error functions not typically addressed in existing models.
>
> > "Making the main body.. more high level..."
>
> Thank you. We will revise the presentation to focus on high-level takeaways in the main paper and move derivations to the appendix. We anticipate using the ICML final version page allowance to improve clarity without altering content.
>
> Once again, we thank you for your detailed feedback and for helping us improve the quality and impact of our work.

---

> > ### Comment · Reviewer_aCoN · 2025-04-05
> >
> > Thank you for your detailed responses! Unfortunately, I think my concerns largely stand. Interestingly, I feel like this paper would be a much better fit for an economics venue than a CS one, partially because of its modeling style and analysis. I do appreciate the effort that the authors put into their rebuttal and discussion of my comments!

---

### Official Review · Reviewer_pD8B · 2025-03-18

**Overall Recommendation:** 2

**Summary:**

The authors propose a model of workforce replacement by AI and run some simulations based on it.

**Claims And Evidence:**

The authors claim to uncover deep truths about the job market, but they rest upon a foundation of assumptions that are not justified. They also do not make any real claims beyond stating they have a working model. This make evaluating the paper difficult as it presents theoretical arguments, but the empirical claims are vague or unusable outside their framework.

**Essential References Not Discussed:**

None come to mind

**Experimental Designs Or Analyses:**

See abov

**Methods And Evaluation Criteria:**

No, they only make claims based on the model. There is not modeling of real labour force dynamics or attempts to test the framework. Just an examination of how it could be applied to an existing dataset. But even when the dataset is used the framework is implicitly enforced thanks to the use of an LLM to generate the data.

**Other Comments Or Suggestions:**

No

**Other Strengths And Weaknesses:**

I am particularly concerned with the assumptions that skills are independent in LLMs and that they will generalize reliably. Both of these are still open questions in the literature, so some discussion of these limitations should at least be included.

**Questions For Authors:**

What am I supposed to gain from the heatmaps? They seem very sensitive to initial conditions/parameter selection. Showing that there are transition points in an econ model is not novel.

**Relation To Broader Scientific Literature:**

If the paper delivered what is promised this might have relevance, but as written does not. I suggest the authors consider either increasing the empirical groundedness of the work before resubmitting to CS-society venue, or do more work to build on the theory and submit to an econ theory venue.

**Theoretical Claims:**

I did not check the theorems. I found the assumptions of independence and lack of connection to the real world to make the actual model irrelevant to my analysis.

---

> ### Author Rebuttal · Authors · 2025-04-01
>
> We thank the reviewer for taking the time to evaluate our submission and for the opportunity to clarify key aspects of our work. Please see this PDF for new figures (link: https://acrobat.adobe.com/id/urn:aaid:sc:eu:fef7d6b2-24f6-4a59-9386-fdf09456ed99).
>
> > "..assumptions that skills are independent.,.."
>
> Our model does **not** assume skill independence. Each skill $s \in [0,1]$ is assigned an ability function, and nearby skills (e.g., 0.7 and 0.8) have similar expected abilities. Thus, correlation across skills is encoded by design. For instance, coding and debugging are correlated due to their proximity in skill space.
>
> The independence assumption applies only to **skill execution**: once ability functions are fixed, realizations of performance across tasks are treated as independent—assuming task inputs are drawn independently. This is standard in modeling both humans and LLMs. We will clarify this in the final version.
>
> Additionally, Sec. 4 (Lines 373L–356R) already considers dependent skill executions: a parameter $p\in[0,1]$ introduces correlation via a shared latent variable $\beta$. We extend Theorem 3.1 to this setting (details appear in response to Reviewer aCoN).
>
> > "deep truths about the job market... but do not make any real claims."
>
> We make no such claim. Rather, we introduce a theoretical framework for analyzing human-AI task-level collaboration, supported by empirical validation. As stated in the abstract and introduction, our contributions include formalizing how job success varies with ability profiles and quantifying the benefit of merging human and AI workers—both supported mathematically and empirically.
>
> We also added new empirical results that may be of interest to practitioners. Notably, we now show that our model captures the **productivity compression** phenomenon reported in Brynjolfsson et al. (2023). Specifically,  Theorem 3.2 implies that merging low-skilled worker $W_1$ with a GenAI tool yields a larger gain in job success probability than merging a high-skilled worker $W_2$, thus narrowing the gap $P_2-P_1$. Since the job success probability correlates with productivity, this explains the empirical narrowing in productivity observed in the cited study.  Our heatmaps (Figure 6 in the PDF) empirically reproduce this effect. To our knowledge, our model is among the first to offer a formal explanation of this compression under realistic ability assumptions (details appear in response to 1zaz.)
>
> > "No modeling of real labour force dynamics or attempts to test the framework."
>
> Our model is not intended as a macroeconomic tool. It is a **task-level, microeconomic framework** that enables reasoning about **skill decomposition** and **human-AI integration**. It is grounded in real-world data: O\*NET (task-skill-job structure), Big-Bench Lite (AI capabilities), and GPT-generated task-skill mappings. Section 4.4 discusses modeling limitations and directions for broader validation.
>
>
> > "I did not check the theorems. I found the assumptions... make the actual model irrelevant."
>
> We respect your decision. However, the theoretical results—such as phase transitions and merging theorems—are central to our work. Understanding when merging yields gains or when success changes abruptly is non-trivial and requires careful analysis. Other reviewers have highlighted the value of this contribution: Reviewer aCoN noted the rigor of the merging results, and Reviewer 1zaz emphasized the model’s potential for interpreting labor studies.
>
> > "What am I supposed to gain from the heatmaps?"
>
> The heatmaps illustrate how job success probability $P$ and merging gain $\Delta$ vary with ability profiles:
>
> - **Phase transition behavior:** The heatmaps confirm whether transitions in $P$ or $\Delta$ occur gradually or abruptly. Distinct color boundaries validate the predicted transitions.
>
> - **Extension beyond theory:** While Theorem 3.2 assumes identical profiles, Figure 4 shows that phase transitions still emerge when profiles differ.
>
> - **Effort required to achieve gains:**
>  The size of the bright region in Figures 2 and 4(b) indicates how easily a large $\Delta$ can be achieved by merging. Figure 2(b) vs. Figure 4(b) shows that merging identical profiles yields more abrupt gains than merging distinct ones.
>
> These visualizations help connect our analytical results to observable behavior in practical scenarios.
>
> > "There is no connection to the real world."
>
> While our model is stylized to allow formal analysis, it is **empirically grounded** and addresses concrete, real-world questions about human-AI integration at the task level. It uses real datasets (O\*NET and Big-Bench Lite) and models skill-action decompositions and merging—challenges at the heart of current research in both ML and labor economics. We hope our clarifications and new results help make this connection clearer.
>
> We thank you again for your time, and hope our responses convey the relevance, rigor, and potential of our work.

---

### Decision · Program_Chairs · 2025-05-01

**Decision:**

Accept (poster)

**Comment:**

This paper presents a mathematical framework for modeling human-AI collaboration in the workplace by decomposing jobs into decision-level and action-level subskills. The authors analyze conditions under which combining agents with complementary strengths improves job success, introduce formal results characterizing phase transitions in job success probability, and demonstrate how the model can explain empirical observations such as productivity compression. The framework is instantiated using the O*NET and Big-Bench Lite datasets, with GPT-generated mappings to calibrate AI and human capabilities.

Overall, reviewers appreciate the proposal of a formal model addressing a timely topic within the growing literature on GenAI in the workforce. The reviewers' primary reservations focus on (1) the gap between the theoretical model and practical applications and (2) whether the paper might be more suitable for economics venues. For the first concern, following reviewer 1zaz’s suggestion, the authors demonstrated in their rebuttal that their model explains the "compression effect" of GenAI tools documented in existing empirical studies. This alleviates some concerns about practical relevance, and I strongly encourage the authors to expand this line of discussion further. Regarding venue fit, while the work has an economics-oriented flavor, I believe the topic is relevant and would attract interest from the ML community. However, I also want to highlight reviewer aCoN's suggestion to clearly convey high-level ML-relevant takeaways in the main text to increase accessibility and appeal to the ML community.